

# Towards ice thickness inversion: an evaluation of global DEMs by ICESat-2 in the glacierized Tibetan Plateau

Wenfeng Chen1,3, Tandong Yao1, Guoqing Zhang1, Fei Li1,3, Guoxiong Zheng2,3, Yushan Zhou1, Fenglin Xu1,3

[1] State Key Laboratory of Tibetan Plateau Earth System Science, Institute of Tibetan Plateau Research, Chinese Academy of Sciences, Beijing 100101, China
[2] Xinjiang Institute of Ecology and Geography, Chinese Academy of Sciences, Urumqi 830011, China
[3] University of Chinese Academy of Sciences, Beijing 100049, China

*Correspondence to*: Wenfeng Chen (chenwf@itpcas.ac.cn)

**Abstract.** Accurate estimates of regional ice thickness, which are generally produced by ice-thickness inversion models, are crucial for assessments of available freshwater resources and sea level rise. Digital elevation model (DEM) derived surface topography of glaciers is a primary data source for such models. However, the scarce in-situ measurements of glacier surface elevation limit the evaluation of DEM uncertainty, and hence its influence on ice-thickness modelling over the glacierized area of the Tibetan Plateau (TP). Here, we examine the performance over the glacierized TP of six widely used and mainly global-

scale DEMs: AW3D30 (30 m), SRTM-GL1 (30 m), NASADEM (30 m), TanDEM-X (90 m), SRTM v4.1 (90 m) and MERIT (90 m) by using ICESat-2 laser altimetry data while considering the effects of glacier dynamics, terrain, and DEM misregistration. The results reveal NASADEM as the best performer, with a small mean error (ME) of –1.0 and a root mean squared error (RMSE) of 12.6 m. A systematic vertical offset existed in AW3D30 (–35.3 ME and 34.9 m RMSE), although it had a similar relative accuracy to NASADEM (~13 m STD). TanDEM-X also performs well (–0.1 ME and 15.1 m RMSE),

but suffers from serious errors and outliers on steep slopes. SRTM-based DEMs (SRTM-GL1, SRTM v4.1, and MERIT) (all ~36 m RMSE) had an inferior performance to NASADEM. However, their errors were reduced in the ablation zone when glacier variations were excluded. Errors in the six DEMs increased from the south-facing to the north-facing aspect and become larger with increasing slope. Misregistration of DEMs relative to ICESat-2 footprint in most glacier areas is small (less than one pixel). An intercomparison of four ice-thickness models: GlabTop2, Open Global Glacier Model (OGGM), Huss-Farinotti

(HF), Ice Thickness Inversion Based on Velocity (ITIBOV), show that GlabTop2 is sensitive to the accuracy of both elevation and slope, while OGGM and HF are less sensitive to DEM quality, and ITIBOV is the most sensitive to slope accuracy. Considering the inconsistency of DEMs acquisition dates, NASADEM would be a best choice for ice-thickness estimates over the TP, followed by AW3D30, and TanDEM-X (if steep and high elevation terrain can be avoided). Our assessment figures out the performances of mainly global DEMs over the glacierized TP. This study not only avails the glacier thickness

estimation with ice thickness inversion models, but also offered references for other cryosphere studies using DEM.



# 1 Introduction

The Tibetan Plateau (TP), which includes the Pamir, Hindu Kush, Karakoram, Himalaya, and Tibet regions, covers an area of ~3 million km2 and has a mean elevation of more than 4000 m a.s.l. (Fig. 1). It accounts for more than 82% of the Earth's land surface area above 4000 m a.s.l. (Fielding et al. 1994), and is often referred to as the Third Pole of Earth or the Asian Water

Tower (Yao et al. 2012) due to its high elevation and abundant water resources in the form of glaciers, snow, permafrost, lakes, and rivers. The TP has a glacierized area of ~$8.3 \times 10^4$ km2 (RGI Consortium, 2017) with an ice volume of ~$6.2 \times 10^3$ km3 (Farinotti et al. 2019), mainly distributed in the Karakoram and Himalaya regions.

Ice thickness is a crucial parameter for assessing the contribution of glaciers to global sea level rise (Kraaijenbrink et al. 2017), quantifying regional water availability (Huss and Hock 2018; Immerzeel et al. 2020), and evaluating cryosphere-related

hazards (Linsbauer et al. 2016; Zheng et al. 2021). In the TP, owing to the lack of in-situ ice thickness measurements (Welty et al. 2020), regional glacier thickness is mainly estimated by ice-thickness inversion models (ITIMs) using open access digital elevation models (DEMs) (Farinotti et al. 2009; Farinotti et al. 2019; Frey et al. 2014). The DEM is a fundamental part of most regional ITIMs (Farinotti et al. 2017), and is often used to determine center flow lines (Maussion et al. 2019), shear stress (Frey et al. 2014; Wu et al. 2020), apparent mass balance (Farinotti et al. 2009), and for ice-thickness interpolation (Huss and

Farinotti 2012). In addition to its use in ITIMs, the DEM has been an essential model input for a wide range of TP glaciology studies, such as glacier inventory (Bhambri et al. 2011; Frey et al. 2012; Ke et al. 2016; Mölg et al. 2018), glacier mass change (Brun et al. 2017; Shean et al. 2020; Zhou et al. 2018), glacier related disasters (Allen et al. 2019; Kääb et al. 2018; Zhang et al. 2019) and projections of glacier or glacial lake evolution (Kaser et al. 2010; Kraaijenbrink et al. 2017; Zheng et al. 2021). The uncertainty in the DEMs can lead to different ITIM outcomes (Frey and Paul 2012; Fujita et al. 2017; Furian et al. 2021;

Kääb 2005), especially for those ITIMs in which the DEM is a crucial input. For example, the sensitivity of the glacier bed topography (GlabTop) model to slope increases for shallower slopes (Paul and Linsbauer 2012), and slope overestimated by ~10% would result in an underestimation of ice thickness of ~32% (Linsbauer et al. 2012). Localized errors and data gaps could affect the estimated ice thickness by 5−25% (Huss and Farinotti 2012). Therefore, it is imperative to choose a suitable DEM source for regional glacier thickness modelling. Farinotti et al. (2017 and 2021) intercompare the performance of most

ITIMs and suggest that consideration of the uncertainty in the input data could improve the model output. However, to our knowledge, the uncertainty in different open access DEMs and its influence on various ITIM outputs over the TP has not been evaluated.

Currently, open-access DEMs covering the whole TP are mainly created by stereo mapping sensors such as ALOS AW3D30 (Tadono et al., 2015), C- or X-band interferometry synthetic aperture radar (InSAR) such as TanDEM-X, and SRTM-C based

products such as NASADEM (Crippen et al. 2016). Shadows and the layover effect of InSAR technology (González and Fernández 2011), along with the deficient orientation of photogrammetrically stereo images (Mukherjee et al. 2013) or low stereo-correlation (Hugonnet et al. 2021) propagated during DEM production, may introduce errors and voids. Filling these voids with other data could result in increased uncertainty (Liu et al. 2019). Additionally, the rugged terrain of glaciers and the





low contrast of snow cover can often lead to geometric distortion and missing data (Reuter et al. 2007; Takaku et al. 2020).

Estimates of the accuracy of DEMs in different terrains and physiognomy, and for different vegetation coverage and land use have been conducted outside the TP using Global Navigation Satellite Systems (GNSS) measurements or high-resolution DEMs (González-Moradas and Viveen 2020; Grohmann 2018; Hawker et al. 2019; Uuemaa et al. 2020). The performance of specific DEMs varied in these studies, indicating that the local terrain and land cover influenced the DEM accuracy. In the TP, glaciers are distributed across different climatic zones and have a wide range of elevations with rugged and complicated terrain

(Fielding et al. 1994; Thompson et al. 2018). GNSS measurements are not accessible for most glaciers, and public access high-resolution DEM is also a limitation due to its long temporal coverage (Shean 2017). The assessment of DEM accuracy in specific regions with limited GNSS measurements and high-resolution DEM is not enough to determine the performance of global DEMs across the whole glacierized TP.

Liu et al. (2019) evaluated the performance of seven public freely-accessed DEMs over the TP with sparse ICESat altimetry

data and suggested that AW3D30 has a high degree of accuracy. However, ICESat data with a footprint of 70 m (larger than the resolution of their estimated DEMs) could result in intra-pixel errors in steep slopes (Uuemaa et al. 2020). Besides, glacial regions were not considered in their studies, due to the variations of glaciers over time. Misregistration among DEMs, which may lead to evaluation bias (Han et al. 2021; Hugonnet et al. 2021; Van Niel et al. 2008), was also neglected. Bearing these issues in mind, and considering the limitations of optics sensors in rugged terrain and the glacier accumulation area (Chen et

al. 2021), it is clear that a further assessment of the performance of AW3D30 is required. Recently, TanDEM-X (released in 2017) and NASADEM (released in 2020) have been reported to have large improvements in accuracy relative to previous DEM products for various land-cover types (Wessel et al. 2018), floodplain sites (Hawker et al. 2019), slightly undulating terrain (Altunel 2019), and mountain environments (Gdulová et al. 2020). Nonetheless, their performance over the rugged and glacierized TP remains unclear.

The purpose of this study is to evaluate the optimal DEM to use for regional ice thickness estimation over the TP. We first evaluated the performance of six widely used DEMs: AW3D30, SRTM-GL1, NASADEM, TanDEM-X, SRTM v4.1, and MERIT which are derived from different sensors and have different resolutions, against ICESat-2 data which has been proven to have a high vertical accuracy and resolution. The elevation differences between these DEMs and the ICESat-2 are systematically analyzed with regard to aspect, slope, elevation, and glacier zones. The influence on the accuracy assessment

of glacier elevation changes, terrain and misregistration among DEMs is then quantified. Finally, we compare the performance of ice thickness estimates derived using the six DEMs against in-situ measurements of ice thickness using Ground Penetrating Radar (GPR). The influence of DEM uncertainties on the model outcomes is also analyzed.



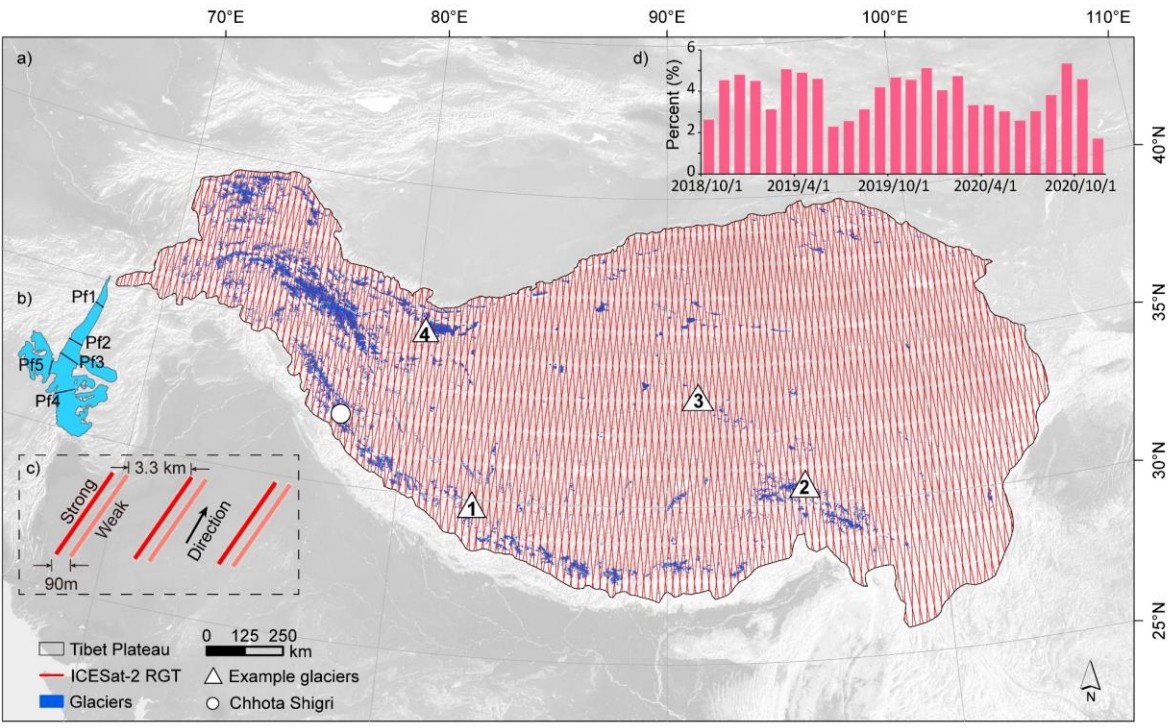

**Figure 1.** Location of the TP and its ICESat-2 reference ground tracks (RGTs). a) ICESat-2 tracks over the TP intersecting with glaciers. The numbered labels refer to glaciers used as examples in Fig.12. b) Location of Ground Penetrating Radar (GPR) profiles over the Chhota Shigri Glacier which is used as an example. c) Relative location of six beams when Advanced Topographic Laser Altimeter System (ATLAS) has backward orientation. Distance between RGTs is 28.8 km. d) Percentage of ICESat-2 data in different months from October 2018 to November 2020. The boundary of the TP is derived from SRTM above 2500 m a.s.l (Zhang et al. 2013).

## 2 Data and Methods

### 2.1 ICESat-2 elevation data referenced

ICESat-2, a follow-on mission to the Ice, Cloud, and land Elevation Satellite (ICESat), was launched on 15 September 2018, with the goal of acquiring Earth's geolocated surface elevation that referenced to the WGS 84 ellipsoid at the photon level. ICESat-2 ATLAS (Advanced Topographic Laser Altimeter System) emits a pulse every 0.7 m along the track covering a horizontal circular area with ~17 m diameter and 0.5 m in vertical extent. We used the ICESat-2 Level-3A land-ice ATL06 product. ATL06 heights are median-based heights derived from a linear-fit model over each segment corrected for first-photon bias. The segment has a length of 40 m centered on reference points at 20-m intervals along the track. The ATL06 product has better than 5 cm height accuracy and better than 13 cm surface measurement precision in the Antarctic (Brunt et al. 2019). The



product also contains land background points. The RGI6.0 glacier inventory (RGI Consortium, 2017) was used to extract

points falling on the glaciers (Fig. 2).

    ICESat-2 ATL06 data covering the TP from October 2018 to November 2020 was downloaded from https://earthdata.nasa.gov/ (Fig. 1). There are 2436 files containing about 100 GB of data in total. The fields: Location (latitude, longitude), surface elevation (h_li), elevation uncertainty (h_li_sigma) and quality (atl06_quality_summary) were used. By combining the quality field (atl06_quality_summary=0) (Smith et al. 2019) with the glacier inventory, a total of 3.5 million

points out of 0.16 billion records over the TP were selected (Fig. 1). The slope, aspect and elevation value of the cell center of the DEMs were extracted for the ICESat-2 footprints.

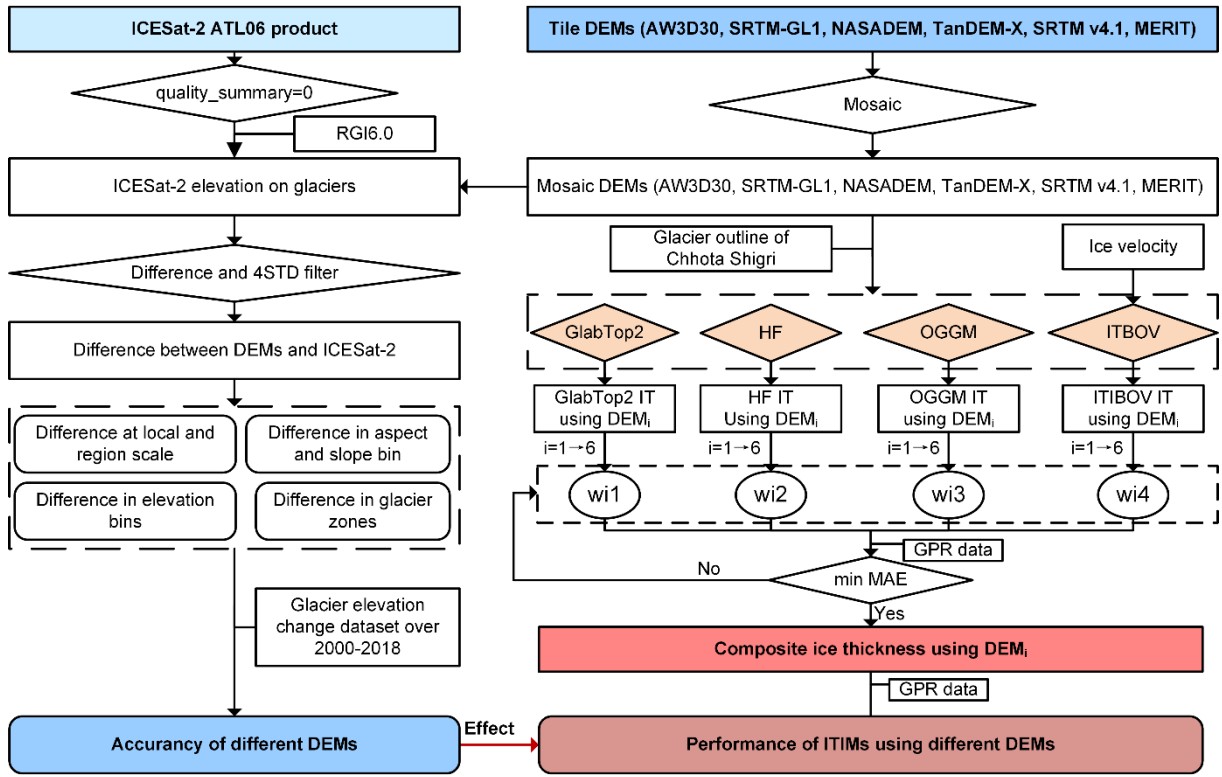

**Figure 2**. Flow chart showing the targets and methods used in this study including accuracy evaluation of DEMs and their effects on ice-

thickness inversion models. The wi1, wi2, wi3 and wi4 denote the weight of each modeled ice thickness, i from 1 to 6 are six different DEMs, and the number 1−4 are the four ice thickness inversion models.

## 2.2 DEMs evaluated (AW3D30, TanDEM-X, NASADEM, SRTM)

Six global-scale DEMs were selected for evaluating their influences on ITIMs, based on popularity, data source, resolution

and sensor type (optics or SAR) (Table 1).



1) ALOS World 3D - 30 m (AW3D30) is acquired by the optics stereo sensor loaded on the Advanced Land Observing Satellite (ALOS) which operated from 2006 to 2011 with a horizontal resolution of 30 m. Data gaps are filled with SRTM, ASTER GDEM v3, ArcticDEM v3, and TanDEM-X 90. Data was acquired from https://www.eorc.jaxa.jp/ALOS/en/aw3d30/index.htm after user registration.

2) TanDEM-X 90 m DEM (hereafter TanDEM-X) is a product derived from the first bistatic X band SAR mission of the world which took place from 2014 to 2016 (Bachmann et al. 2021). It is a pixel-reduced product of the global TanDEM-X DEM with a pixel spacing of 0.4 arcseconds (12 m). The official reported absolute vertical and horizontal accuracy is better than 10 m at the 90% confidence level. It is noted that the current release is a non-edited version: areas with outliers, noise and voids remain. The original data was collected during different seasons and years, and the influence of ablation and

accumulation of glaciers should also be noted. Data was acquired from https://download.geoservice.dlr.de/TDM90/.

3) NASADEM is a new product released in 2020, which is derived by reprocessing the original SRTM signal data using updated interferometric unwrapping algorithms and auxiliary data, such as ICESat, to reduce voids and improve vertical accuracy (Crippen et al. 2016). Remnant voids are filled mainly by Global Digital Elevation Model (GDEM) v3 data. This data was downloaded from https://search.earthdata.nasa.gov/.

4) Other SRTM based DEMs (SRTM-GL1, SRTM v4.1, MERIT). SRTM-GL1 (30 m) is an extensively used DEM in ITIMs. The first open-access ice-thickness database of global glaciers also adopted SRTM-GL1 as its DEM source (Farinotti et al. 2019). Voids were primarily filled by ASTER GDEM2. SRTM v4.1 and MERIT were selected to compare with TanDEM-X, and simultaneously estimate the influence of DEM resolution on ITIMs. SRTM v4.1, with a spatial resolution of 90 m, is produced by the method proposed by Reuter et al. (2007), including merging tiles, filling small

holes iteratively and interpolating across the holes using a range of methods, according to the size of hole, and the land type surrounding it (https://cgiarcsi.community/data/srtm-90m-digital-elevation-database-v4-1/). SRTM v4.1 was also used to compare against the performance of SRTM-GL1 to estimate the influence of resolution. MERIT is also widely used with a spatial resolution of 90 m. It was developed by removing absolute bias, stripe noise, speckle noise, and tree height bias from the existing spaceborne DEMs (SRTM3 v2.1 and AW3D30 v1) using multiple satellite datasets and

filtering techniques (Yamazaki et al. 2017). Its accuracy was significantly improved, especially in flat regions (Yamazaki et al. 2017). The overall accuracy is similar to TanDEM-X in floodplain sites (Hawker et al. 2019), but lower in short vegetation. The dataset was downloaded from http://hydro.iis.u-tokyo.ac.jp/~yamadai/MERIT_DEM/.





**Table 1** Details of the selected DEMs.

| Item | Version | Acquisition time | Release time | Resolution (m) | Sensor type | Description | Source |
|---|---|---|---|---|---|---|---|
| AW3D30 | v2.2 | 2006–2011 | Apr. 2019 | 30 | Optical | Generated from its original version processed at 5 m or 2.5 m grid spacing. Voids were filled with other open-access DSMs such as SRTM, ASTER GDEM, ArcticDEM, etc. | https://www.eorc.jaxa.jp/ALOS/en/aw3d30/index.htm |
| SRTM-GL1 | v003 | Feb. 2000 | Sep. 2015 | 30 | SAR C-band | ASTER GDEM2, USGS GMTED2010 or USGS National Elevation Dataset were used for voids filling | https://earthexplorer.usgs.gov/ |
| NASADEM | v1 | Feb. 2000 | Feb. 2020 | 30 | SAR C-band | Reprocessing of the original SRTM radar signal data and telemetry data with updated algorithms and auxiliary data such as ASTER GDEM2, ICESat, AW3D30 | https://search.earthdata.nasa.gov/ |
| TanDEM-X | v1.0 (Non-edited version) | 2010–2015 | Feb. 2019 | 90 | SAR X-band | A product variant of the 12 m (0.4 arcsec) DEM product in version 1.0 from the world's first bistatic SAR mission | https://download.geoservice.dlr.de/TDM90/ |
| SRTM v4.1 | v4.1 | Feb. 2000 | Nov. 2018 | 90 | SAR C-band | Reproduced using the method of Reuter et al. (2007). | https://drive.google.com/drive/folders/0B_J08t5spvd8RWRmYmtFa2puZEE |
| MERIT | v1 | Feb. 2000 | Oct. 2018 | 90 | SAR C-band | Improved by removing multiple error components from the existing spaceborne DEMs (SRTM3 v2.1 and AW3D-30m v1). | http://hydro.iis.u-tokyo.ac.jp/~yamadai/MERIT_DEM/ |



### 2.3 Ice thickness inversion models

Tiles of six DEMs (AW3D30, TanDEM-X, NASADEM, SRTM-GL1, SRTM v4.1, and MERIT) were used to form a mosaic of terrain data covering the whole TP. Four ice-thickness inversion models (GlabTop2, HF, OGGM, ITBOV) were used. The Chhota Shigri Glacier located in western Himalaya for which the GPR data were available (Fig.1) was selected as an example to evaluate the influence of DEM uncertainty on the ITIMs. Full details of the ITIMs are given below:

GlabTop (Glacier bed topography) is based on the theory that glacier thickness is mainly determined by the slope of the terrain (Linsbauer et al. 2012; Linsbauer et al. 2009; Paul and Linsbauer 2012). It is assumed that the glacier is an ideal plastic fluid,
with bottom slip being ignored. Based on the empirical relationship between mean shear stress along the centerlines and the range of glacier elevation (Haeberli and Hoelzle 1995) (Eq. 1), the actual basal shear stress $\tau$ can be determined.

$$\tau = 0.005 + 1.598\Delta H - 0.435\Delta H^2$$

$$\tau = 150kPa, if\ \Delta H > 1600 \tag{1}$$

where $\Delta H$ is the elevation range of glacier. The ice thickness $h$ can then be determined from Eq. (2)

$$h = \frac{\tau}{f \rho g \sin \alpha} \tag{2}$$

where $f$ is the shape factor, $\rho$ is glacier density ($850\pm60$ kg/m$^3$) (Huss 2013), $g$ is the acceleration due to gravity (9.8 m/s$^2$) and $\alpha$ is the slope. Glabtop2 is an automated method for calculating ice thickness, similar to GlabTop, but avoiding digitizing the
branch lines. For details refer to Frey et al. (2014).

HF (Huss-Fainotti) model is based on the mass balance principle which relates the surface mass balance of the glacier ($b$) to the ice flux and variation in the glacier thickness. Given the ice flux, ice thickness can be calculated according to Glen's ice flow law (Farinotti et al. 2009a; Huss and Farinotti 2012).

$$h = \sqrt[n+2]{\frac{q(1-fsl)}{2A} \frac{n+2}{\left(f \rho g \sin \alpha\right)^n}} \tag{3}$$

where $h$ is the mean elevation of band thickness, $q$ is the ice flux, $fsl$ is the basal slip correction factor, $n=3$ is the exponent
of flow law, $\rho$ is glacier density ($850\pm60$ kg/m$^3$) (Huss 2013), $g$ is the acceleration due to gravity (9.8 m/s$^2$), $f$ is the valley shape factor (0.8) (Cuffey and Paterson 2010) and $A$ is the Glen flow rate factor ($3.24\times10^{24}$ Pa$^{-3}$ s$^{-1}$) (Cuffey and Paterson 2010; Gantayat et al. 2014).

This method defines a new variable $\tilde{b} = \dot{b} - \rho g \frac{\partial h}{\partial t}$, where $\tilde{b}$ is the apparent mass balance, $\dot{b}$ is the glacier surface mass balance,

and $\frac{\partial h}{\partial t}$ is the glacier surface elevation change. $\tilde{b}$ is linearly related to the elevation and has nothing to do with whether or not
the glacier is in a stable state. In the absence of mass balance data and thickness change data on the surface of a glacier, the





ice flux $q$ can be obtained by estimating $\tilde{b}$ , which is determined from experience (Huss and Farinotti 2012). Ice thickness in each elevation band can then be determined by substituting into Equation (3). Finally, $h$ is extrapolated, in combination with the slope, to obtain the distributed ice thickness, according to the parameters in Huss and Farinotti (2012).

The Open Global Glacier Model (OGGM) is based on the same concept as HF, but has two main differences (Maussion et al. 2019). Firstly, the method described in Kienholz et al. (2014) is used to automatically obtain the middle streamlines and watershed division. Secondly, the apparent mass balance data are reconstructed from the local climatic dataset from variables such as precipitation and temperature.

The Ice Thickness Inversion Based On Velocity (ITIBOV) model obtains the ice thickness by combining the surface velocity field with the Glen ice flow law (Gantayat et al. 2014; Glen 1955; McNabb et al. 2012):

$$h = \frac{n+1}{2A} \frac{(1-k)u_s}{\left(f \rho g \sin \alpha\right)^n} \tag{4}$$

where $h$ is ice thickness, $u_s$ is glacier surface velocity, and $k$ is the contribution ratio of basal slip velocity relative to the $u_s$. We used the ITSLIVE dataset (Gardner 2019 ) as the $u_s$ input. We assumed that basal slip only occurred during the warm seasons, and $k$ was calculated by dividing the annual glacier velocity by winter glacier velocity (Wu et al. 2020). Data from the Global Land Ice Velocity Extraction from Landsat 8 (GoLIVE) dataset with a date separation length of less than 96 days are used to estimate the monthly velocity (Fahnestock et al. 2016; Scambos 2016), allowing the winter velocity (December,

January and February) and annual mean velocity to be calculated. Basal factor $k$ was calculated as 0.80 (Fig. S1). Some shared parameters, such as creep factor, shape factor and basal creep factor are the same in all four models.

It is possible that an ensemble of the output from different models can improve the modeled thickness (Farinotti et al. 2017; Farinotti et al. 2021). Therefore, after calculating the ice thickness from four models using different DEMs, we calculated an ensemble ice thickness using the same DEM but with different models. The weighting given to each model is iteratively

calculated to achieve a minimal mean absolute error (Fig. 2).

**2.4 Accuracy assessment**

The error in the DEMs is considered to be the difference between the DEM elevation and the ICESat-2 measurement. To remove the influence of outliers, elevation differences outside four standard deviations were removed. Mean error (ME), mean absolute error (MAE), median error, root mean square error (RMSE), standard deviation (STD), and normalized median

absolute deviation (NMAD) were calculated for the error assessments. NMAD and ME were used to assess the disturbance from extreme errors (Gdulová et al. 2020).

$$ME = \frac{1}{n} \sum_{1}^{n} (H_{ICESat\text{-}2} - H_{DEM}) \tag{5}$$

$$MAE = \frac{1}{n} \sum_{1}^{n} abs(H_{ICESat\text{-}2} - H_{DEM}) \tag{6}$$





$$RMSE = \sqrt{\frac{1}{n}\sum_{1}^{n}(H_{ICESat-2} - H_{DEM})^2} \qquad (7)$$

$$STD = \sqrt{\frac{1}{n-1}\sum_{1}^{n}(H_{ICESat-2} - H_{DEM} - ME)^2} \qquad (8)$$

$$NMAD = 1.4826 * median(abs(H_{ICESat-2} - H_{DEM})) \qquad (9)$$

## 3. Results

### 3.1 Accuracy of DEMs

Figure 3 shows a comparison of elevation from the six DEMs with the ICESat-2 data. The four standard deviations (that is 4

std) filter on the differences between ICESat-2 and DEMs used to filter out extreme outliers excluded less than 1% of the data.
Overall, non-irregular deviation existed among these DEMs and ICESat-2 elevation after filtering. The ICESat-2 vs DEMs
values are distributed tightly around the fit line with a slope coefficient close to 1, with no obvious differences among the R2.
NASADEM and TanDEM-X performed the best in terms of intercept and fit RMSE, with very little difference to the ICESat-
2 data. For the other four DEMs, there are obvious systematic shifts which can be inferred from the high R2 values, but high

intercept values.

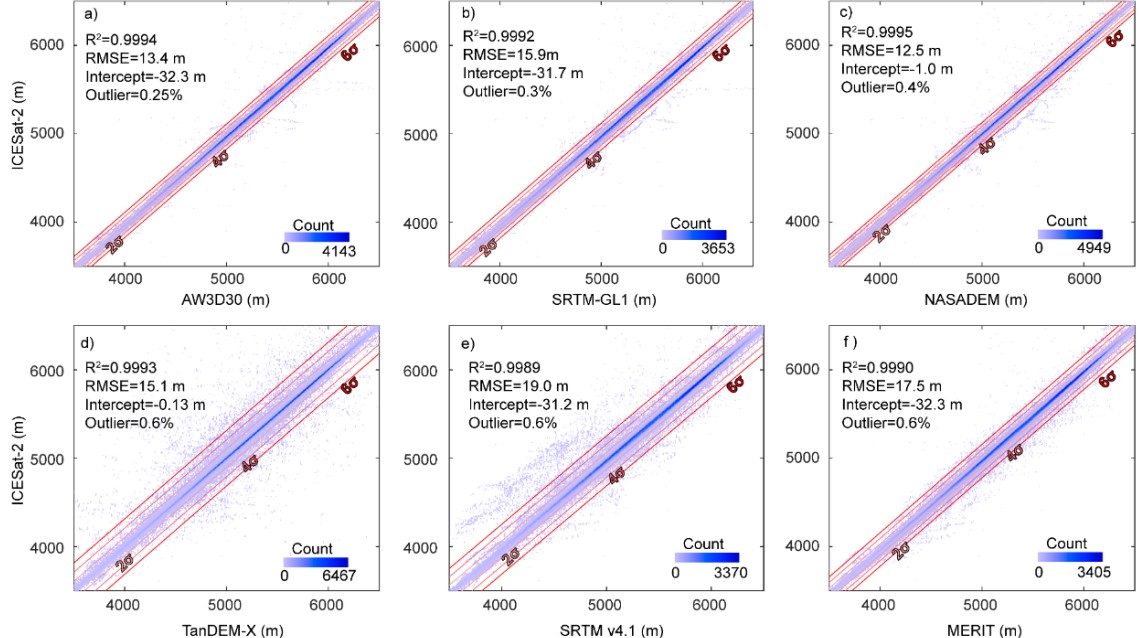

**Figure 3**. Differences between six DEMs and ICESat-2 elevation. a) AW3D30, b) SRTM-GL1, c) NASADEM, d) TanDEM-X, e) SRTM
v4.1, and f) MERIT. The gradually lighter red lines denote the range within 2, 4 and 6 std of the mean. The text at the top left of each panel



gives the fit results for data within 4 std of the mean. 'Outlier' denotes the proportion of outliers relative to the total records. 'R2', 'RMSE',

and 'Intercept' are fit results when the slope coefficient is set to 1. Elevation range was cut to 3500−6500 m, the range in which most elevations values are located, to show clearly the effect of using different multiples of the std from the mean.

The difference statistics for the six DEMs are presented in Figure 4. Statistically, Median and ME differed little, and extreme values did not influence the ME much after the 4 std filter was applied. STD was slightly larger than NMAD, especially for

TanDEM-X, indicating larger discrepancies (Höhle and Höhle 2009). NASADEM performed far better than the other two 30-m resolution DEMs, with smaller RMSE (12.6 m), MAE (9.4 m), and ME (–1.0 m). AW3D30 has a lower absolute accuracy (RMSE: 34.9 m, ME: –32.3 m), but a similar relative accuracy to NASADEM because of the similar overall dispersion (~13 m) and spatial scale. SRTM-GL1 and NASADEM are both produced from same original SAR data, but have large differences in RMSE (35.5 vs 12.6 m), MAE (31.9 vs 9.4 m), and ME (–31.7 vs –1.0 m). The new algorithm and auxiliary data applied in

NASADEM do indeed greatly improve the absolute accuracy of the product over glacierized terrain. The quality of TanDEM-X was the best out of the 90-m resolution DEMs with small RMSE (15.1 m), MAE (8.9 m), ME (–0.1 m), and STD (15.1 m). SRTM v4.1 and MERIT are both error-reduced products from SRTM3 v2 (Reuter et al. 2007; Yamazaki et al. 2017), and they have similar behavior, with ME about –32 m and RMSE ~37 m.

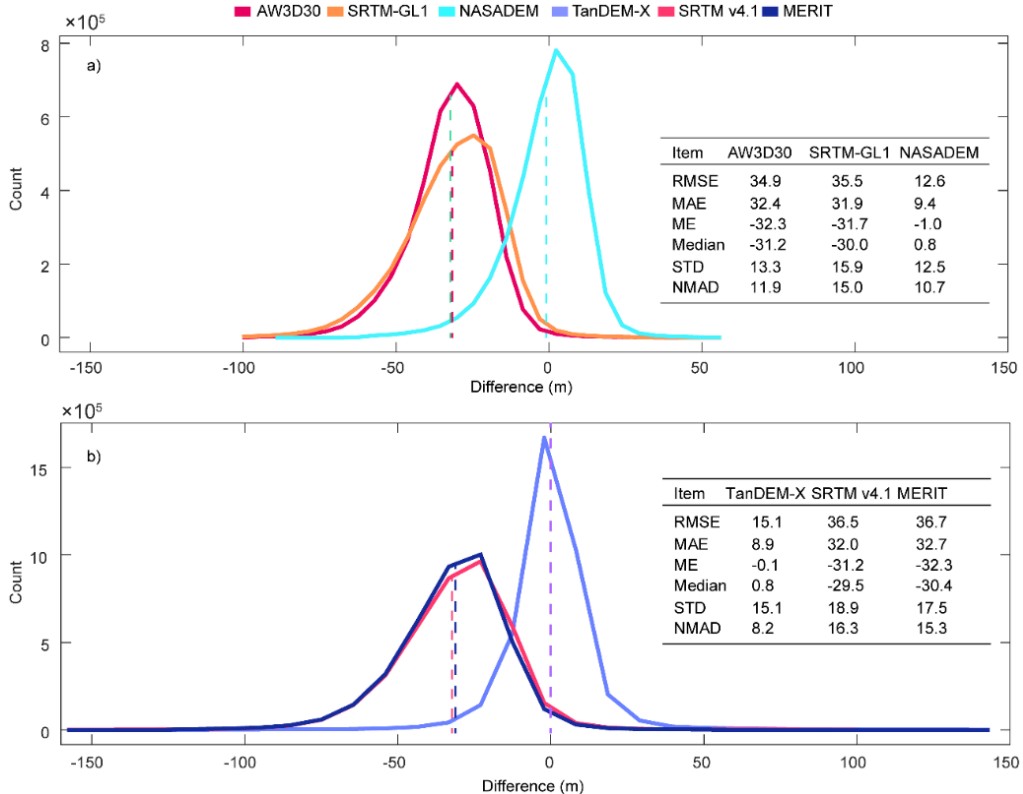





**Figure 4.** Overall difference (m) statistics between six DMEs and ICESat-2 elevation. a) 30-m-resolution DEMs, AW3D30, SRTM-GL1 and NASADEM. b) 90-m-resolution DEMs, TanDEM-X, SRTM v4.1 and MERIT.

The spatial distribution of the ME and STD are shown in Figure 5. AW3D30, SRTMGL-1, SRTM v4.1 and MERIT all have large negative ME over the TP, while the ME of NASADEM and TanDEM-X are mostly within the range –10 to 10 m. For
these two DEMs, the ME in the Himalaya is more negative than that in southeast Tibet, and it is slightly positive in western Kunlun and the Karakoram mountains. It is worth noting that in the Himalaya and southeast Tibet, the ME of NASADEM is more negative than that of TanDEM-X. Overall, STD along the Hindu Kush-Himalaya and southeast Tibet was larger than in other regions. Thereinto, STD in southeast Tibet was relatively larger (>12 m). Specifically, the STD of AW3D30 and NASADEM was minimum and spatially relevant. TanDEM-X performs well in overall statistics (Fig. 4b), but it is not stable.
Spatially, it performed worse than SRTM-GL1 in terms of STD (Fig.5b). The STD and ME of SRTM v4.1 and MERIT are almost the same in space (Fig.5b), corresponding to their similar overall STD (both ~18 m) and ME (both ~32 m) values (Fig.4b).

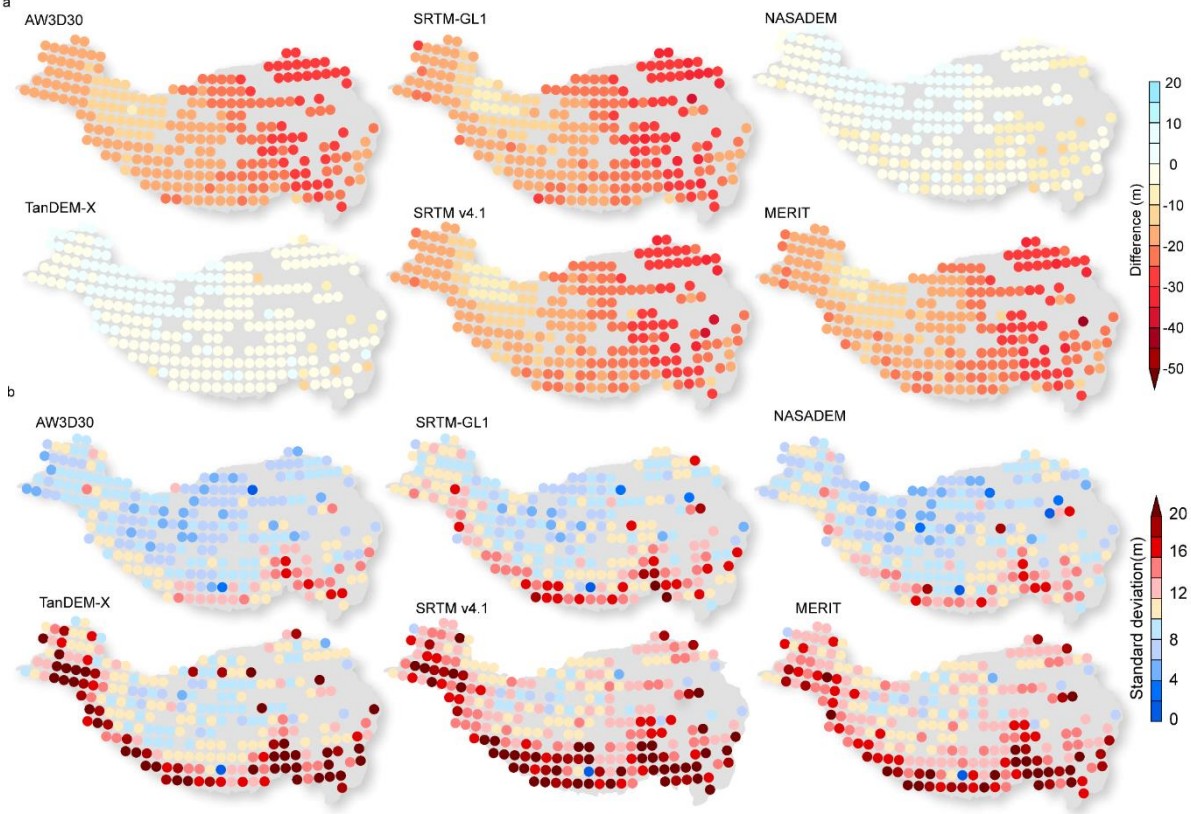






**Figure 5.** Aggregated spatial mean error (ME) (a) and standard deviation (STD) (b) between six DEMs and ICESat-2 elevation for 1°×1° cells across the TP.

### 3.2 Differences between DEMs and ICESat-2 in aspect, slope and elevation

The influence of terrain factors on the differences between the DEM and ICEsat-2 elevations for the six DEMs are presented
in Figure 6. The influence of aspect is most apparent for SRTM-GL1, with a median value of about –25 m in the south aspect which increased in magnitude gradually towards to the north aspect (–35 m). A similar pattern, but with a smaller amplitude (±5 m) is apparent for the NASADEM and TanDEM-X (Fig.6a).

For NASADEM and TanDEM-X, the differences plotted against slope are distributed around zero, with mean median values of about –1.6 and –1.0 m, respectively (Fig.6b). For the other DEMs, the differences with slope are mainly less than zero, with
a median range of –30 to –48 m and a mean upper quartile of ~20 m. The median differences of the 30-m DEMs generally increased along the slope. However, for the 90-m DEMs, the difference increased with slope at first, but then decreased on steep slopes. For all DEMs, the spreads of differences become larger as the slope becomes steeper. This increase is most obvious for TanDEM-X and SRTM v4.1, with rates of 0.71 m/degree ($r$=0.96, $p$<0.01) and 0.60 m/degree ($r$=0.83, $p$<0.01). On slopes of less than 20°, TanDEM-X has the best quality with a mean median value of 0.2±0.39 m and 5.8±2.2 m, but
increased disparity on steeper slopes. Overall, relative to the other DEMs, NASADEM behaves best against slope in terms of spread and median value. MERIT shows a slight advantage over SRTM v4.1 with a reduced spread for steep slopes.

The differences for all DEMs generally increased with elevation, with fluctuations at very high elevations (Fig.6c). For NASADEM and TanDEM-X, the differences are negative at lower elevations and slightly positive at higher elevations; NASADEM varied from –70 to 20 m over the full elevation range, and from –10 to 10 m over the range 4500−6500 m, where
measurements are concentrated; TanDEM-X varied from –15 to 20 m over the full elevation range, and from around –5 to 5 m between 4500 and 6500 m. For the other four DEMs, the differences all remained below zero for the full range of elevations. The SRTM-GL1, SRTM v4.1 and MERIT differences changed similarly from –100 to 0 m. In comparison, the differences for AW3D30 were smaller for lower elevation bins, ranging from –70 to 0 m.





**Figure 6.** Differences between six DEMs and ICESat-2 with terrain factors. (a) 5° aspect bin. (b) 2° slope bin. (c) 200 m elevation bin. (d) Percentage (%) of data in each aspect, slope and elevation bin.



### 3.3 Differences between DEMs and ICESat-2 in different glacier zones

Differences in different glacier zones were also estimated and are shown in Figure 7a-d. We divided it into four sub-zones
using the maximal, median and minimum elevation from the RGI glacier inventory (Fig. 7e). Here we consider Zone 1 to be
the ablation area and Zone 4 the accumulation area. Zone 2 and Zone 3 are transition areas. Crests of the probability distribution
of differences located in the negative axis range in Fig. 7a move to the right in Fig. 7b−d. Correspondingly, ME, MAE and
RMSE all decrease when moving from Zone 1 (ablation area) to Zone 2 (transition area) (Fig.7 and Table S1). Spatially, areas
in the glacier terminus are subject to more melting (Brun et al. 2017) leading to this decrease. The ME of the SRTM based
products SRTM-GL1, SRTM v4.1, NASADEM and MERIT decreased by 8.7, 7.3, 8.1 and 7.8 m, respectively. Temporally,
the values of the DEM acquired in earlier periods decreased more. The ME decreased by a larger value (5.6 m) for AW3D30,
which was acquired in 2006−2011, than for that of TanDEM-X (3.5 m), which was acquired in 2010−2015.

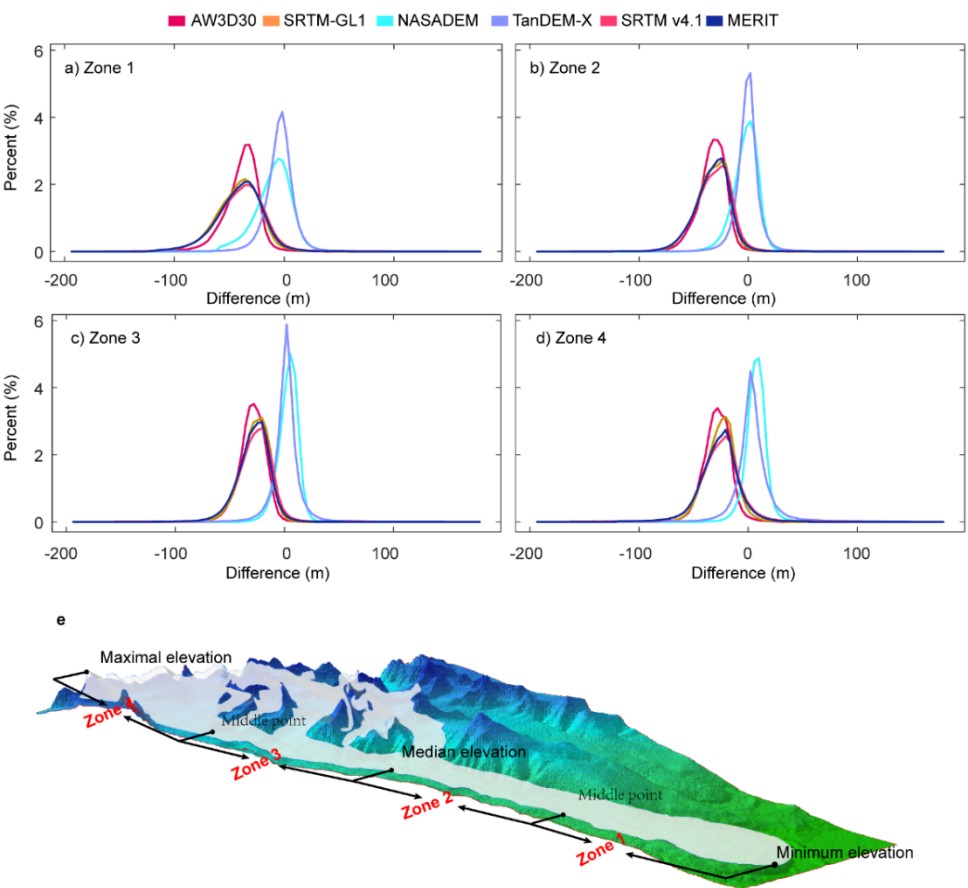

**Figure 7.** Probability distribution of the difference between six DEMs elevation and ICESat-2 elevation in different glacier zones (a−d) that
defined in panel (e).





ME, MAE and RMSE in Zone 3 and Zone 4, near or in the accumulation area, are almost all smaller than the corresponding values in Zone 1 and Zone 2 (Fig. 7 and Table S1). For TanDEM-X and NASADEM, which have better absolute accuracy than the other DEMs, ME changed to positive values in Zone 3 and Zone 4. Usually, in the accumulation area, glaciers have a

positive or less negative elevation change (Li and Lin 2017; Maurer et al. 2019; Rankl and Braun 2016), therefore, accumulation may be concerned with changes in Zone3 and Zone 4.

In terms of STD, NASADEM performed best in all glacier zones, except for Zone 1, with values ranging from 9.2 to 16.5 m (Table S1). AW3D30 had the best performance of all DEMs in Zone 1, and was the next best performer overall, with a STD varying from 12.0 to 20.7 m. The STD of TanDEM-X was better than that of SRTM-GL1 in Zone 1 and Zone 2, but worse in

Zone 3 and Zone 4 where it suffered from discrepancies. MERIT showed slight improvements in STD at ~2% relative to SRTM v4.1.

## 3.4 Comparisons of ice thickness modelled by DEMs

The models are not adjusted independently according to the difference between the output and GPR results. Therefore, the results are not indicators of the performance of the models but rather references for examining the influence of different DEMs

on specific ITIMs. The effect of the DEMs on the model outcomes are presented in Figure 8 and are quite obvious. Mean ice thickness differs, according to the DEM used, by up to 88%, 4%, 47% and 7% for GlabTop2, HF, ITIBOV and OGGM, respectively. The deepest ice thickness differs by up to 55%, 25%, 16% and 18% for GlabTop2, HF, ITIBOV and OGGM, respectively.

The mean ice thicknesses from GlabTop2 and ITIBOV using the 90-m DEMs (that is TanDEM-X, SRTM v4.1 and MERIT)

are ~30 m less than those obtained from using 30-m DEMs (that is AW3D, SRTM-GL1 and NASADEM) (Fig. 8). GlabTop2 and OGGM using AW3D30, and HF and ITIBOV using NASADEM output the maximal mean thickness. GlabTop2, ITIBOV and OGGM using TanDEM-X, and HF using SRTM-GL1 output the minimum mean thickness.

The influence of different DEMs on ITIMs can also be identified when making a comparison with the GPR results (Fig. 8 and Table 2). If median error is used as the criteria, GlabTop2 using NASADEM, HF using AW3D30, ITIBOV using NASADEM and OGGM using NASADEM achieved the relatively best simulation (Fig. 9). If RMSE was used, GlabTop2 using

NASADEM, HF using SRTM-GL1, ITIBOV using NASADEM and OGGM using TanDEM-X performed best (Table 2).

In different glacier zones, each DEM-model combination has its merits and weakness (Table 2). Totals of 9, 3, 4, 3 and 1 output achieved the minimum RMSE in profiles by different ITIMs using AW3D30, SRTM-GL1, NASADEM, TanDEM-X and MERIT, respectively. Overall, NASADEM, as input to GlabTop2 and ITBOV, performs best, with RMSE values of 75.4

and 61.3 m, respectively; SRTM-GL1 performed the best in HF with an RMSE of 50 m; TanDEM-X performed the best in OGGM with an RMSE of 52.8 m (Table 2).
**Figure 8.** Distribution of modelled ice thickness of Chhota Shigri Glacier (location shown in Fig.1) using AW3D30, SRTM-GL1, TanDEM-X, SRTM v4.1, NASADEM and MERIT. (a) Glabtop2; (b)HF; (c) ITBOV; (d) OGGM; (e) composite result. Mean (ME) and maximum (MAX) modelled ice thickness are given in each panel.

Following the procedure of Farinotti et al. (2017), results from the four models are further composed to achieve the minimum MAE between the modelled and GPR thicknesses (Fig. 8e). The weights for each model in ten experiments are shown in Table



S2. After composition, the mean thickness using different DEMs ranged from 77 (acquired based on TanDEM-X) to 91 m

(acquired based on NASADEM). The thickness error of the results based on NASADEM is best (median value 2.3 m), followed

by TanDEM-X (median value –7.5 m). The minimum RMSE is for AW3D30 (45.9 m), followed by NASADEM with a RMSE

of 47.7 m.

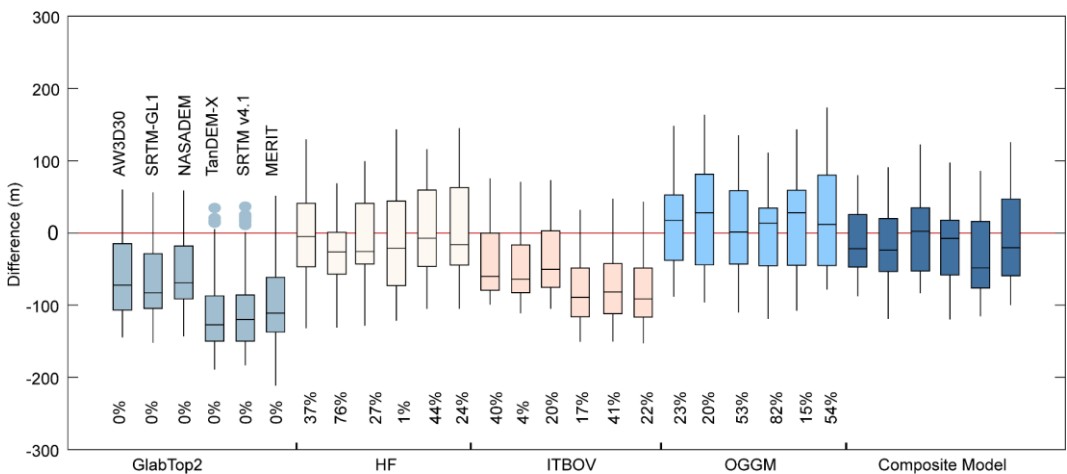

**Figure 9.** Point-by-point deviation comparison between the modelled and measured ice thickness from GlabTop2, HF, ITIBOV, OGGM

and the composite result. In each group, the boxes are plotted in the order: AW3D30, SRTM-GL1, NASADEM, TanDEM-X, SRTM v4.1

and MERIT. Different models using the same DEM are aggregated by weights (labeled at the bottom) to achieve minimum mean absolute

error.








**Table 2** RMSE (m) of modelled ice thickness compared with ground penetrating radar (GPR) measurements on each profile. Bold numbers denote the best model performance on each profile.

| Model | DEM | No. of ground penetrating radar profiles | | | | | |
| --- | --- | --- | --- | --- | --- | --- | --- |
| | | 1 | 2 | 3 | 4 | 5 | All |
| GlabTop2 | AW3D30 | 56.6 | **84.5** | **45.4** | 102. | 103. | 79.7 |
| | SRTMG-GL1 | **54.8** | 94.6 | 62.4 | 104. | 90.6 | 83.3 |
| | NASADEM | 54.9 | 77.7 | 50.2 | **100.** | *87.4* | **75.4** |
| | TanDEM-X | 117.1 | 144. | 97.4 | 139. | 127. | 124. |
| | SRTM v4.1 | 101. | 133. | 93.2 | 132. | 135. | 118.7 |
| | MERIT | 110.0 | 120. | 70.9 | 154. | 132. | 118.1 |
| HF | AW3D30 | **27.2** | 29.8 | 83.7 | 30.7 | 92.5 | 61.6 |
| | SRTM-GL1 | 28.4 | 58.4 | **33.3** | **30.6** | 88.1 | **50.0** |
| | NASADEM | 37.0 | 29.2 | 69.0 | 36.6 | 86.9 | 56.2 |
| | TanDEM-X | 45.5 | **27.8** | 91.4 | 76.0 | 85.0 | 72.4 |
| | SRTM v4.1 | 49.3 | 36.4 | 74.9 | 44.8 | 86.4 | 61.3 |
| | MERIT | 51.4 | 33.6 | 87.6 | 42.4 | **86.2** | 65.8 |
| ITBOV | AW3D30 | **69.3** | 61.9 | **45.6** | **66.4** | 70.8 | 61.4 |
| | SRTMG-GL1 | 70.7 | 71.7 | 52.6 | 70.6 | 61.4 | 64.8 |
| | NASADEM | 71.3 | **53.8** | 56.7 | 67.2 | **60.5** | **61.3** |
| | TanDEM-X | 104. | 116.6 | 55.5 | 84.0 | 110.4 | 91.8 |
| | SRTM v4.1 | 102. | 114.5 | 51.8 | 73.4 | 111.3 | 88.3 |
| | MERIT | 108. | 117.6 | 54.8 | 80.4 | 112.7 | 91.9 |
| OGGM | AW3D30 | **32.0** | 47.2 | 80.5 | **62.5** | **34.6** | 58.9 |
| | SRTMG-GL1 | 33.5 | 56.9 | 101. | 65.1 | 41.1 | 70.2 |
| | NASADEM | 33.9 | 57.6 | 84.5 | 68.8 | 40.7 | 64.3 |
| | TanDEM-X | 33.0 | **41.2** | **59.4** | 65.2 | 47.2 | **52.8** |
| | SRTM v4.1 | 37.4 | 46.1 | 81.4 | 68.7 | 39.9 | 61.6 |
| | MERIT | 38.3 | 52.1 | 104. | 53.8 | 49.7 | 69.7 |
| Composite | AW3D30 | 43.8 | 30.8 | **43.4** | **41.5** | 71.1 | **45.9** |
| | SRTMG-GL1 | **33.3** | 47.8 | 47.7 | 46.7 | 80.0 | 52.1 |
| | NASADEM | 52.5 | 27.8 | 43.9 | 54.6 | 60.6 | 47.7 |
| | TanDEM-X | 50.6 | 32.2 | 65.6 | 66.5 | **59.4** | 57.7 |
| | SRTM v4.1 | 46.9 | **26.5** | 47.5 | 69.0 | 59.6 | 51.9 |
| | MERIT | 57.0 | 27.7 | 76.3 | 53.0 | 72.9 | 60.7 |


## 4. Discussion

### 4.1 Factors related to the differences of DEMs: glacier elevation change, terrain

The identified extreme outliers (Fig. 3) are mostly located in the glacier terminus, high elevation and high slope regions (Fig. 10a−b). Extreme glacier melt, such as in southeastern Tibet, and surges, as observed in the Karakoram, can also lead to dramatic



elevation changes, resulting in large difference (Fig. 10c). This glacier elevation change effect is also reflected in the spatial
distribution of difference (Fig. 5), elevation bins (Fig. 6c) and glacier zones (Fig. 7). The differences at lower elevations are
negative, and generally increase with elevation, consistent with the fact that glaciers melt at lower elevations and accumulate
at higher elevations (Cuffey and Paterson 2010). The differences in the NASADEM and TanDEM-X data with elevation
comply with these features. NASADEM was acquired in 2000 and TanDEM-X was acquired in 2010−2015, and the value of
NASADEM is more negative than TanDEM-X, as would be expected. By making a comparison between SRTM-GL1 and
NASADEM from the same original data, we conclude that the negative differences of the other four DEMs through the
elevation bins may be related to absolute vertical shift. MERIT shows less improvement over SRTM v4.1 in glacierized terrain
than in the flat regions in terms of both absolute and relative accuracy (Yamazaki et al. 2017). The relatively more negative
and larger values of ME and STD along the Hindu Kush-Himalaya, southern Tibet (Fig. 5) and in glacier zones (Fig. 6) are
also related to glacier elevation change.

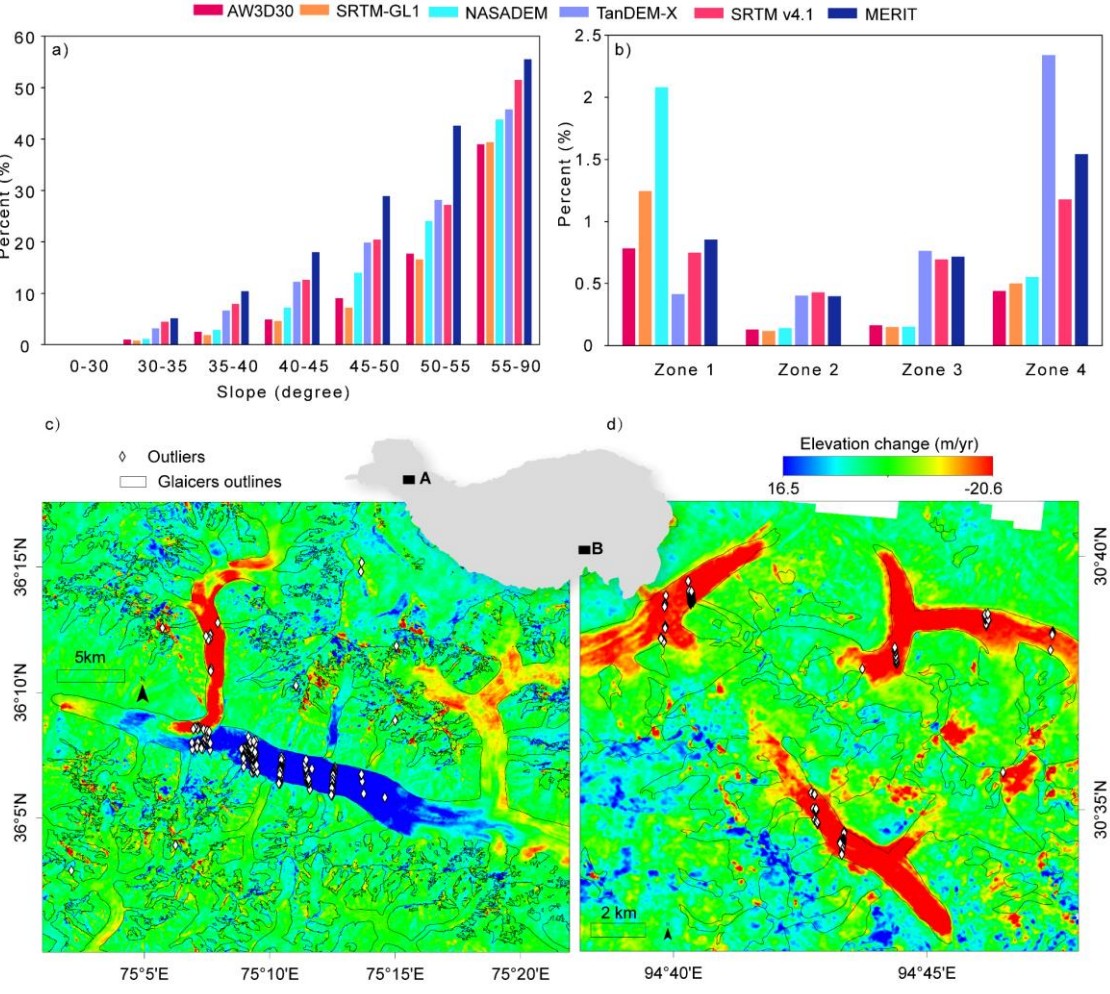





**Figure 10.** Distribuion of excluded extreme outliers. Proportion of outliers accounting for total number in slope bins (a) and each glacier Zone (b). Examples of locations of excluded points overlaid with glacier surface elevation change in Karakoram (c) and southern TP (d). Locations of these two examples are labeled A and B in the central insert. Glacier elevation change data covering 2000−2019 is from Shean

et al. (2020).

**Table 3** Comparisons of differences between four SRTM based DEMs and ICESat-2 elevation over glacier zones before and after adjustment. ICESat-2 data acquired in February are used to calcuate the differences. Glacier zones are defined according to Fig. 8e.

| Item | Zone | Before (m) | | | | After (m) | | | |
|---|---|---|---|---|---|---|---|---|---|
| | | SRTM-GL1 | NASADEM | SRTM v4.1 | MERIT | SRTM-GL1 | NASADEM | SRTM v4.1 | MERIT |
| Mean | 1 | −44.6 | −12.4 | −43.9 | −44.4 | −31.5 | 0.1 | −30.1 | −31.1 |
| error | 2 | −32.6 | −2.9 | −31.7 | −32.6 | −26.2 | 3.4 | −25.3 | −26.0 |
| | 3 | −27.2 | 3.0 | −28.0 | −28.5 | −23.6 | 6.6 | −24.0 | −24.6 |
| | 4 | −25.5 | 7.0 | −26.9 | −26.6 | −24.1 | 8.1 | −26.4 | −26.1 |
| Absolute | 1 | 44.9 | 16.9 | 44.6 | 44.9 | 31.7 | 7.6 | 30.8 | 31.5 |
| mean error | 2 | 32.7 | 9.0 | 32.1 | 32.8 | 26.4 | 7.8 | 25.8 | 26.4 |
| | 3 | 27.4 | 7.5 | 28.7 | 28.8 | 24.1 | 9.4 | 25.1 | 25.2 |
| | 4 | 26.1 | 9.2 | 29.3 | 28.2 | 25.3 | 11.4 | 30.0 | 28.6 |
| Standard | 1 | 20.8 | 18.0 | 24.0 | 22.6 | 13.6 | 10.2 | 17.0 | 16.0 |
| deviation | 2 | 14.7 | 11.7 | 16.7 | 15.7 | 12.7 | 9.7 | 14.7 | 14.1 |
| | 3 | 12.9 | 9.0 | 17.2 | 15.2 | 14.1 | 10.3 | 17.4 | 15.7 |
| | 4 | 13.8 | 9.5 | 23.1 | 19.5 | 16.2 | 11.8 | 26.5 | 23.5 |
| RMSE | 1 | 49.2 | 21.9 | 50.0 | 49.9 | 34.3 | 10.2 | 34.6 | 34.9 |
| | 2 | 35.7 | 12.1 | 35.9 | 36.2 | 29.1 | 10.3 | 29.3 | 29.6 |
| | 3 | 30.1 | 9.4 | 32.8 | 32.3 | 27.5 | 12.2 | 29.6 | 29.2 |
| | 4 | 29.0 | 11.7 | 35.5 | 33.0 | 29.0 | 14.3 | 37.4 | 35.2 |


After error correction, using the glacier elevation change dataset covering 2000−2019 (Shean et al. 2019). The mean difference in Zone 1 decreased sharply by ~13 m for the SRTM based DEMs. The ME of NASADEM reached as low as 0.1 m in Zone 1. Improvements are also obvious along the elevation profiles of the four glaciers selected across the TP shown in Fig. 11. Before correction, the ICESat-2 elevation is lower than DEMs elevation in Fig. 11a−c and higher in Fig. 11d. After

correction, the ICESat-2 and NASADEM profiles nearly overlap. However, similar improvements are not obvious in Zone 3





and Zone 4. This may be related to the slight elevation change in the accumulation region (Brun et al. 2017; Shean et al. 2020), and high uncertainty due to steeper slopes and higher elevations (Fig. 6b−c)

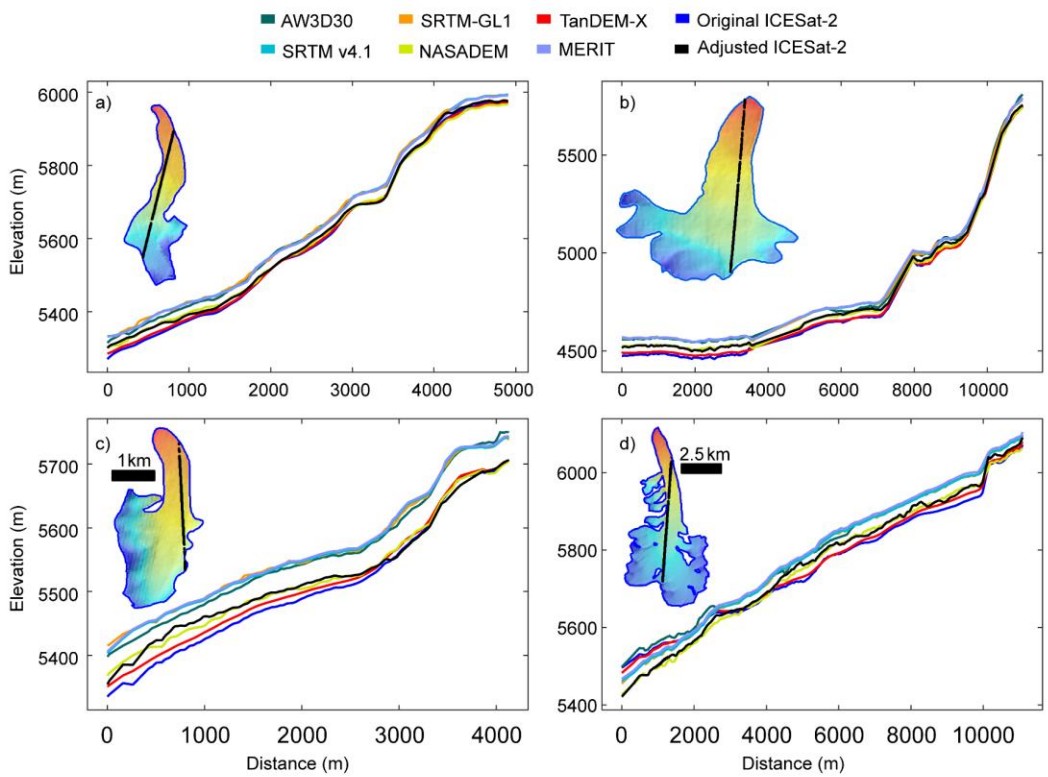

**Figure 11.** Elevation profiles of the original and adjusted six DEMs and ICESat-2 along the ICESat-2 tracks on four selected glaciers across the study region. Locations of glaciers in a−d are labelled 1−4 in Fig. 1, respectively.


ICESat-2 data covering the period October 2018 to October 2020 repeat every 91 days. Therefore, variations of ICESat-2 elevation data caused by glacier fluctuations will have influenced the error statistics (Fig. 12a). Precipitation on the TP mainly occurs in June−August (Maussion et al. 2014). Hence, after precipitation accumulation on glaciers in spring and summer, the

elevation have increased, and the mean difference decreased. With little accumulation, the glacier melt and sublimate in autumn and winter (Li et al. 2018), meaning decreased elevation, and an increase in the mean difference. However, the magnitude of these changes is much smaller, at a level of less than 3 m (Fig. 12), compared with the large ME, MAE and RMSE magnitude of most of the DEMs (with the exceptions of TanDEM-X and NASADEM) (Fig. 4). When taking all points from different seasons into consideration, the seasonal effects could also partly cancel each other out. If only the ICESat-2 data from February

was used (Table 3), NASADEM and TanDEM-X still perform better than others. Therefore, we conclude that the fluctuations of ICESat-2 data have no influence on the performance of the DEMs. Additionally, the atmospheric forward scattering and





subsurface scattering of ICESat-2 photons, which are not quantified in ATL06, may also lead to a biased estimate of elevation (Smith et al. 2019).



**Figure 12.** Influence on elevation differences between ICESat-2 and six DEMs from glacier elevation change and terrain factors. (a) Mean absolute difference between six DEMs and ICESat-2 in different seasons during 2018−2020. The spring, summer, autumn, and winter are defined as March−May, June−August, September−November and December−February, respectively. The histogram at the bottom shows the percentage of the total number of points in each season. (b) Examples of elevation and shaded relief of six DEMs in the Shisha Pangma region.


The elevation differences have a strong dependence on terrain factors (Fig. 7a−b). The differences with aspect show

contrasting features to the distribution of measurements in different aspects (Fig. 7d). The largest errors are concentrated in

the north aspect, as was also reported in previous studies (Gorokhovich and Voustianiouk 2006; Shortridge and Messina 2011),

in which they were attributed to the orientation of the sensor (Gdulová et al. 2020; Shortridge and Messina 2011). However,

here, the data from different sensors all show this aspect dependence, and we infer that it may be related to the accordant

distribution of data in different slopes with aspect. There are many more measurements with steeper slopes in the north aspect,

and less measurements with flatter slopes in the south aspect (Fig. 13). The error and spread become larger with steeper slopes

(Fig. 7b), as also reported by Liu et al. (2019) and Uuemaa et al. (2020), maybe due to geometric deformation or shadow (Liu

et al. 2019) . Therefore, the error variation with aspect tends to be related to steeper slopes (Gdulová et al. 2020; Gorokhovich

and Voustianiouk 2006).

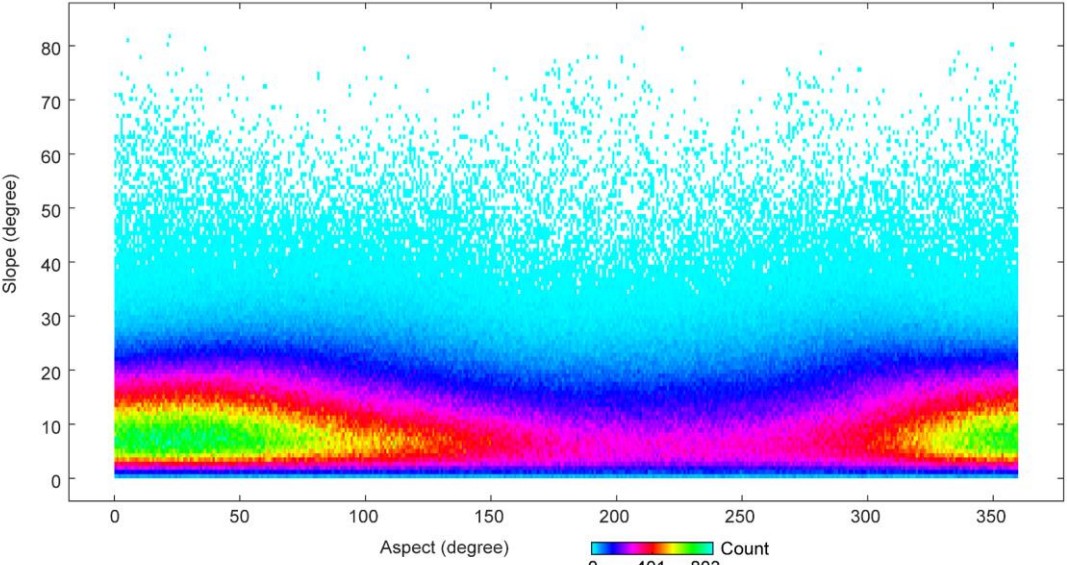

**Figure 13.** Distribution of measurements in different slopes with aspect.

Almost half the points in the 55°−90° slope region are identified as extreme outliers (Fig. 10a). Differences also show large

discrepancies for all DEMs in the steeply sloping regions where voids and large errors are frequent (Falorni 2005). Steep slopes

combined with low resolution led to variations in the spread of differences in Fig. 6b. Spreads of differences were larger on

steep slopes for the 90-m DEMs than those of the 30-m DEMs. Intra-pixel variation aggravates this effect in steeply sloping

regions (Uuemaa et al. 2020), lower resolution or reduced pixel DEMs smooth the terrain details and lead to inaccurate

elevation compared with the 20-m footprint of ICESat-2 points. The spread and the number of outliers gradually increased

with the slope, especially for the TanDEM-X case (Fig. 7b). Using the terrain in the rugged Shishapangma region (Fig. 12b)



as an example, we can see that the elevation from TanDEM-X suffers from serious errors along the ridge at high elevations with the output almost blurred. Even so, TanDEM-X still has overall accuracy advantages over SRTM v4.1 and MERIT, indicating the high quality of TanDEM-X in low relief regions (Fig. 7b).

## 4.2 Influence of misregistration

Misregistration among DEMs, which has been ignored in previous research (González-Moradas and Viveen 2020; Liu et al. 2019), is an important error source when looking at DEM differences (Hugonnet et al. 2021; Van Niel et al. 2008). This study intends to give direct insights about the quality of uncorrected DEM products, so the misregistration problem was not tackled before the evaluations were carried out. However, the influence of misregistration was evaluated. According to the sinusoidal relationship between aspect and error differences between two DEMs (Van Niel et al. 2008), using the co-registration method

in Nuth and Kääb (2011) and ICESat-2 points outside the glaciers, offset pixels at the 1°×1° grid scale were estimated (Fig. 14).

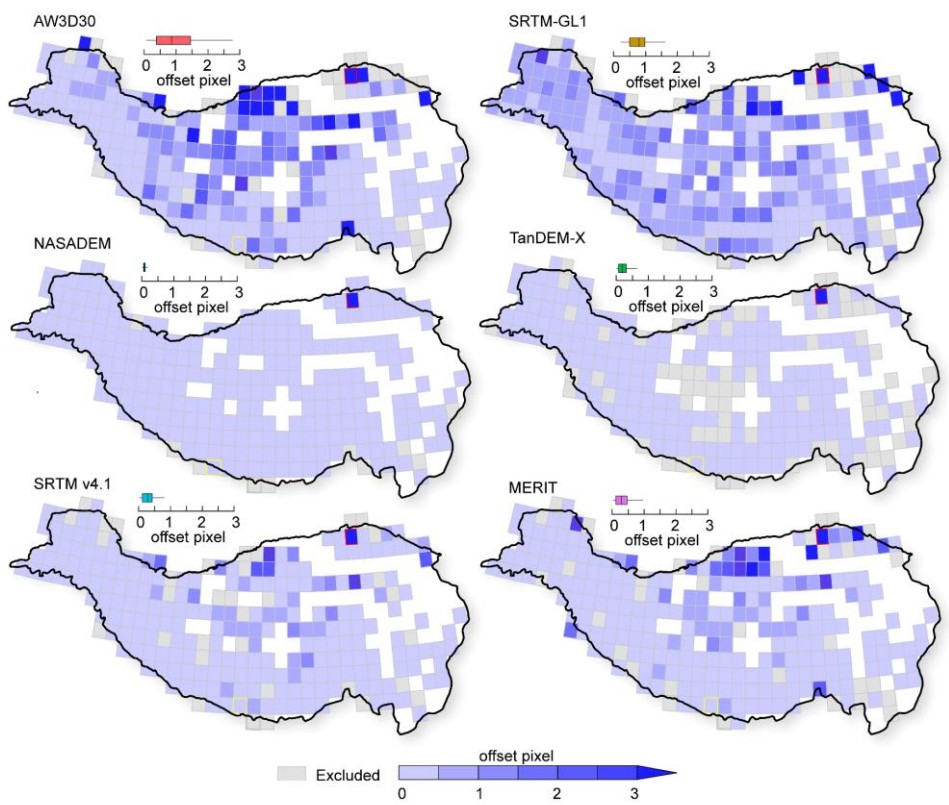

**Figure 14.** Offset pixels of DEMs relative to ICESat-2 on a 1°×1° grid. Only grid squares with $R^2$ greater than 0.5 and number greater than 1000 are displayed. Insert at the top of each figure is the distribution of offset pixels.






Misregistration was found to be worst in eastern Kunlun Mountains and Qilian Mountains where only a small number of glaciers developed (Fig. 1) and was slight in the south of the TP. All the DEMs mismatch by less than one pixel, except for AW3D30, which had the worst misregistration in the north and inner TP (Fig. 14). Considering that only the cell center value was used, sub-pixel shift may have little influence. A comparison of errors before and after correcting sub-pixel misregistration

confirms this conclusion (Fig. 15b). The probability distribution of difference before and after co-registration was almost the same, as were ME, MAE, STD and RMSE (Table S2). However, examples of pixel misregistration strongly affected the probability distribution, except for AW3D30, SRTM-GL1, SRTM v4.1 and MERIT (Fig. 15a); STD changed by 1−3 m, while ME, MAE and RMSE changed by less than 1.2 m (Table S2). The probability distribution symmetry varies. Hence, we supposed that the symmetry variations of difference compensate the effect of offset. Since glaciers are distributed among the

mountains with different aspects, if we shift the DEM in the x- and y-directions, the increased differences would be compensated for by decreased elevation differences (Fig. 15c). Therefore, the large errors remaining in these four DEMs should be due to systematic deviations (Han et al. 2021), rather than the influence of misregistration.

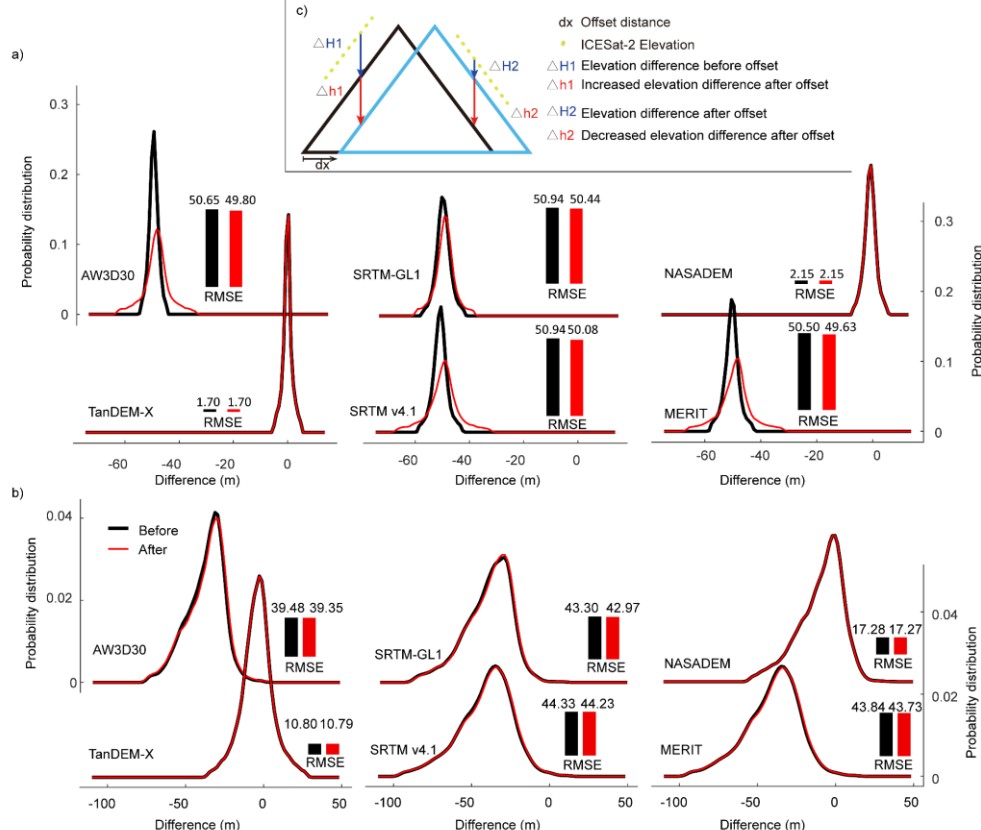

**Figure 15.** Comparison of elevation difference distribution between six DEMs and ICESat-2 before and after co-registration. (a) Several-
pixel scale marked in red and yellow in Fig. 16a and b, respectively; (b) Sub-pixel scale; (c) Concept of influence of DEM registration on elevation difference.





### 4.3 Influence of DEMs on ice thickness estimated by ITIMs


Different DEMs resulted in differences in ITIM ice thickness with a range of 32−65% (Fig. 10). Generally, the outcome with GlabTop2 and ITBOV using 30-m DEMs is better than with the 90-m DEMs, with error differences of 52% and 45%, respectively. This is different from the conclusion that improving only DEM resolution, without calibrating the shape factor, did not benefit the model result in GlabTop2 (Ramsankaran et al. 2018). But when we used a calibrated shape factor of 0.66 as suggested by Ramsankaran et al. (2018), the model results from the 30-m DEMs were still better than those from the 90-m DEMs (Fig. 16a). With GlabTop2, elevation data was used to determine not only the slope, but also the shear stress (Frey et


al. 2014). A +5º error in slope caused more than a –34.1 % difference in the output for slopes of less than 20º. Additionally, relative elevation errors had an enormous impact (Fig. 16 c). For glaciers with an elevation range of less than 400 m, which accounted for 41% of the total number and 5% of the total area, +10, +30, and +50 m errors in elevation range caused more than +2%, +6% and +10% differences in output. Such errors in elevation range had greater influence (Fig. 19b), especially for small glaciers, which have smaller elevation ranges. These two errors propagate and lead to a much larger overall error (Table


3). Thus, GlatbTop2 with NASADEM and AW3D30 as the best quantity input achieved the best RMSE in comparison with GPR measurements. In contrast to the other ITIMs, the ITIBOV model directly estimated the ice thickness at each grid cell according to cell velocity information without interpolation. The slope sensitivity of ITIBOV is higher than that of GlabTop2, with a +5º error in slope causing more than a –71.4% difference in the output for slopes of less than 20º (Fig. 16b). The flatter the slope, the more sensitive ITIBOV is to the slope error (Fig. 16b). Although along- and across-track slope data are provided


in the ICESat-2 ATL06 product, they are incompatible with the slope estimated from DEMs due to their different data formats and algorithms used (Burrough and McDonell 1998; Smith et al. 2019). Moreover, the surface terrain of glaciers changes with time due to accumulation, melting and motion (Dehecq et al. 2018; Shean et al. 2020). Nevertheless, the accuracy of the DEMs estimated here could also provide some information about slope accuracy. The better relative accuracy of NASADEM and AW3D30 means that ITIBOV with these DEMs as input led to the relatively best outcomes (Table 2).


For HF and OGGM, the modelled results did not show large differences when 30-m DEMs were replaced with 90-m resolution DEMs (Fig. 9): means that high spatial resolution improved the outcome little (Pelto et al. 2020). For the HF model, elevation data was used for convergence calculation of apparent mass balance and mean slope in elevation bins (Farinotti et al. 2009; Farinotti et al. 2019), whereas, for OGGM, it is used to extract flowlines, shear stress at flowlines and mass balance at an elevation (Maussion et al. 2019). The relative accuracy of DEMs was more vital than absolute accuracy for these two


models. Although NASADEM and TanDEM-X had the large advantage of absolute accuracy, the output of HF and OGGM using these two DEMs did not have much advantage over that using the other DEMs (Fig. 9). The STD of RMSE values for HF and OGGM using six DEMs are 7.0 and 6.1 m, respectively (Table 2). STD of mean ice thickness by HF and OGGM using six DEMs are 1.1 and 1.9 m (Table 2). The HF and OGGM models are not very sensitive to the DEM absolute accuracy. The performance of AW3D30 in OGGM and SRTM-GL1 in HF is even slightly better than NASADEM in these two models (Table




2). Specifically, the better performance of SRTM-GL1 should be attributed to the calculation of slope. Though the model has
high sensitivity to the slope (Fig. 16a), the mean slope in each elevation band was used, defined as a tangent of the width and
elevation difference in the elevation bin (Huss and Farinotti 2012).

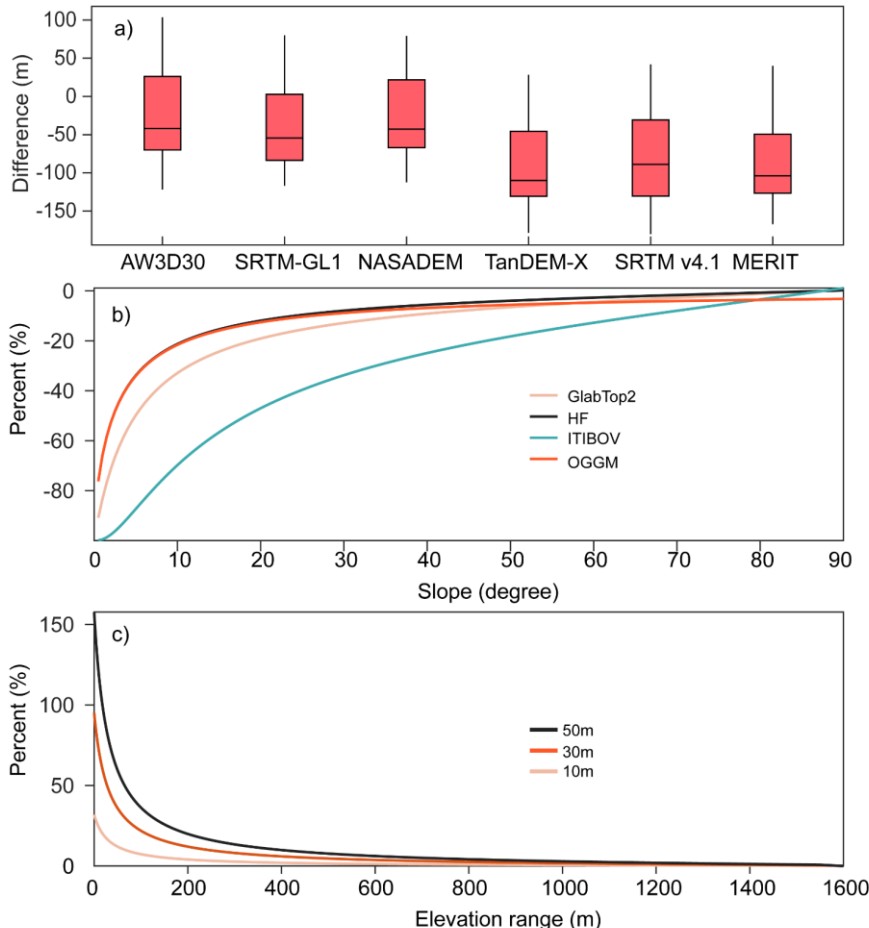

**Figure 16.** Sensitivity test of shape factor, slope and elevation on ice-thickness inversion models. (a) Difference between modelled thickness
and GPR measurements when a calibrated shape factor of 0.66 was used in GlabTop2; (b) Percentage difference of modelled ice thickness
from GlabTop2, HF, ITIBOV and OGGM when there is +5° slope error; (c) Percentage difference of modelled ice thickness from GlabTop2
when the elevation range error is +10, +30 and +50 m for different elevation ranges.

The different models have various levels of robustness to the quality of the input DEMs. When the models are
comprehensively utilized, the influence of the input DEM manifests itself (Fig. 9 and Table 2). The RMSE of ITIMs from 30-
m DEMs was 16.8% less than that from 90-m DEMs. Models using AW3D30 and NASADEM, equipped with higher
resolution and better relative accuracy, achieved the best outcomes. However, it should be noted that the large misregistration
in AW3D30 in the northern TP may lead to the mismatch between terrain and glacier outlines. This will lead to an





overestimation of slope and a consequential underestimation of ice thickness (Huss and Farinotti 2012), due to the mountain
terrain being relatively steeper than the glaciers.

**5. Conclusions**

In the present study, six DEMs (that is AW3D30, SRTM-GL1, NASADEM, TanDEM-X, SRTM-GL1 and MERIT) from
different sensors with different spatial resolutions were evaluated using ICESat-2 data. The influence of glacier dynamics,
terrain and misregistration on the DEM accuracy were analyzed. Out of the three 30-m DEMs, NASADEM was the best
performer with a small ME of –1.0 and a RMSE of 12.6 m. Out of the three 90-m DEMs, TanDEM-X performed best with an
ME of –0.1 and a RMSE of 15.1 m. The quality of TanDEM-X was stable and unprecedented on shallow slopes, but suffered
from serious problems on steep slopes, especially along the steep ridges. For AW3D30, a systematic vertical and horizontal
offset existed on glacier terrain, however, it still has a similar relative accuracy to NASADEM. SRTM-based DEMs (that is
SRTM-GL1, SRTM v4.1 and MERIT) (~36 m RMSE) were inferior to NASADEM, although, when glacier variations were
excluded, all of their errors were reduced in the ablation zone. MERIT shows little improvement over SRTM v4.1 in glacierized
terrain. The influence of glacier elevation change on the elevation difference is larger for DEMs acquired earlier, at low
elevations and in the ablation region. However, this does not influence the conclusion that NASADEM performed the best,
followed by TanDEM-X. For all the DEMs, the errors increased from the south-aspect slope to north-aspect slope, controlled
by the increasing error with slope. Misregistration errors in the glacier-rich region are mostly within one pixel, benefiting from
the 20 m footprint of ICESat-2, relative to the 30- or 90-m resolution DEMs, and only have a small influence on the evaluation.

The effect of DEM accuracy on ice-thickness inversion models depends on the model properties. Generally, a higher
resolution DEM was helpful for better model outcomes due to the intra-pixel influence. For the widely used GlabTop2 model,
which is sensitive to the accuracy of elevation and slope, using NASADEM, with the highest absolute accuracy, as the input,
facilitated the best outcome. Although the OGGM and HF models are less sensitive to the quality of DEM, the use of
NASADEM or AW3D30, both with a high relative accuracy, was still favorable. Among the four ice-thickness inversion
models, ITIBOV was the most sensitive to slope accuracy. Ice-thickness inversion models using AW3D30 or NASADEM as
input gave the best outcomes. These two DEMs also perform the best when four ice-thickness inversion results were aggregated
by the minimum MAE optimization method to form an ensemble.

Considering the influence of inconsistency in data acquisition time on generating glacier terrain, we suggest that NASADEM
is the best choice for ice-thickness inversion models over the TP. AW3D30 could be a good substitute if its systematic shift
was corrected. TanDEM-X is an appropriate alternative for glaciological research focusing on the flat glacier terminus, but it
requires further improvement for use in steep terrain or for ice-thickness inversion.



**Code/Data availability**

IceSAT-2 ATL 06 is acquired at https://nsidc.org/data/atl06; AW3D30 is acquired at https://www.eorc.jaxa.jp/ALOS/en/aw
3d30/index.htm; SRTM-Gl1 is acquired at https://earthexplorer.usgs.gov/; NASADEM is acquired at https://search.earthdata
.nasa.gov/ ; TanDEM-X 90m is acquired at https://download.geoservice.dlr.de/TDM90/; SRTM v4.1 is acquired at https://dr
ive.google.com/drive/folders/0B_J08t5spvd8RWRmYmtFa2puZEE; MERIT is acquired at http://hydro.iis.u-tokyo.ac.jp/~ya
madai/MERIT_DEM/; Glacier elevation change data is acquire at https://zenodo.org/record/3600624.

**Author contribution**

W.F Chen, T.D Yao and GQ. Zhang designed the outline of this study. W.F Chen processed the data and make all the figures.
Fei Li estimate the ice thickness using OGGM. All authors contributed to writing the paper.

**Competing interests**

The authors declare that they have no conflict of interest.

**Acknowledgements**

This study was supported by grants from the Second Tibetan Plateau Scientific Expedition and Research (STEP) program
(2019QZKK0201), the Strategic Priority Research Program (A) of the Chinese cademy of Sciences (XDA20060201) and the
Natural Science Foundation of China (41871056).

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
