# Peer review of "Towards ice thickness inversion: an evaluation of global DEMs in the glacierized Tibetan Plateau"

_The Cryosphere, 2021_

## Referee Comment (RC1)

735

740

[referee-annotated manuscript omitted]

---

## Referee Comment (RC2)

**Comments on**

**"Towards ice thickness inversion: an evaluation of global DEMs by ICESat-2 in the glacierized Tibetan Plateau" by Wenfeng Chen et al.**

**Referee #2**

**Overview**

This paper by Chen et al. presents a method to evaluate existing regional scale DEMs using the recently available ICESat-2 elevation product. The DEMs are then applied to model the ice thickness of glaciers in the Tibetan Plateau region. The quality of the inversion results is then analyzed to prove the effectiveness of the ICESat-2 based evaluation. The paper is generally well structured. It showed that the ICESat-2 data provided a comparison dataset for selecting an optimal DEM that can be used as input for ice thickness inversion. This work should be useful to researchers of The Cryosphere community. The paper needs to be revised according to the following major and minor comments.

The manuscript needs a serious improvement of both formal English writing and scientific meaning.

**Major comments**

1. The subtitles in the Data and Methods sections need to be improved to reflect the actual contents and be logic (*ICESat-2 elevation data referenced*, *DEMs evaluated*, ......).

2. The ICESat-2 data were used to evaluate DEMs. It should be discussed how the DEMs were generated, including data sources, time periods, and uncertainties. During the time differences between the ICESat-2 data and MEDs there may be glacier surface changes that may affect the evaluation. If this is not considered, would this also affect the thickness inversion results?

3. ICESat-2 Level-3A land-ice ATL06 product was used in this work. There should be a good understanding of the quality of this product itself, although a systematic cal-val may not have been performed in the glacierized Tibetan Plateau region. I would like to see even a general discussion in that regard. I suggest to add some relevant references in the Data section:

Brunt, K. M., Neumann, T. A., and Smith, B. E.: Assessment of ICESat-2 Ice Sheet Surface Heights, Based on Comparisons Over the Interior of the Antarctic Ice Sheet, Geophys. Res. Lett., 46, 13072–13078, https://doi.org/10.1029/2019GL084886, 2019.

Brunt, K. M., Smith, B. E., Sutterley, T. C., Kurtz, N. T., and Neumann, T. A.: Comparisons of Satellite and Airborne Altimetry With Ground-Based Data From the Interior of the Antarctic Ice Sheet, Geophys Res Lett, 48, e2020GL090572 https://doi.org/10.1029/2020GL090572, 2021.

Li, R., Li, H., Hao, T., Qiao, G., Cui, H., He, Y., Hai, G., Xie, H., Cheng, Y., and Li, B.: Assessment of ICESat-2 ice surface elevations over the Chinese Antarctic Research Expedition (CHINARE) route, East Antarctica, based on coordinated multi-sensor observations, The Cryosphere, 15, 3083–3099, https://doi.org/10.5194/tc-15-3083-2021, 2021.

Elevation differences between crossovers formed by ICESat-2 tracks may be used for the elevation accuracy evaluation.

As ATL06 product comes from ATL03 photons grouped in 40 m segments, would ATL03 data provide more terrain details? This may be outside of the scope of this paper. However, a discussion of this potential would be helpful to the readers in their future research.

4. Based on your results, the accuracy of ICESat-2 data is better than the compared global-scale DEMs. Have you considered to use the ICESat-2 data to improve the quality of the DEMs, especially in areas where ICESat-2 along and across track data are available. Again, this may be considered as a future work.

**Minor comments**

**Page 1, Line 11:** Replace "derived" with "derived from".

**Page 1, Lines 12–14:** This sentence is awkwardly phrased. You may rephrase by separating it in to two sentences.

**Page 1, Line 17:** Please be clear if it is the horizontal or vertical accuracy.

**Page 1, Lines 23-24:** Change "one pixel" to one grid spacing.

**Page 1, Lines 24-28:** These sentences are not communicating well. Please cut them to short and simple sentences.

**Page 2, Lines 33 & 36:** Change "km2 ", "km3 " to "km$^2$", "km$^3$".

**Page 3, Line 87 and Page 4, Lines 107-108:** When you mention ICESat-2 in these two places, please mention the cal-val efforts and introduce the most recent accuracy assessment results by Brunt et al. (2019, 2021) and Li et al. (2021).

**Page 4, Line 95:** Replace "intersecting" with "covering".

**Page 4, Line 105:** ~17 m diameter is the design value. This value needs to be updated according to the new study: Magruder, L. A., Brunt, K. M., and Alonzo, M.: Early ICESat-2 on-orbit geolocation validation using ground-based corner cube retro-reflectors, Remote Sensing, 12, 3653, https://doi.org/10.3390/rs12213653, 2020.

**Page 4, Line 106:** In addition to first-photon bias, transmit-pulse shape correction should be mentioned also.

**Page 5, Line 115:** The ATL06 data provide slopes in along and across tracks. They are derived at different scales (along track with denser points and cross track with fewer points and longer separations). Have you considered the difference between these two types of slopes?

**Figure 2:** "Difference in aspect and slope bins".

**Figure 2:** Make different blue boxes into just blue color.

**Page 6, Line 127:** Please specify how severe is the gap situation and the effectiveness of gap filling.

**Page 6, Line 135:** All website links need to add the last visit date to ensure their availability.

**Page 8, Lines 160-162:** The sentence is unclear. Please rewrite.

**Page 10, Line 209:** Please justify your choice of 4-std of differences between ICESat-2 and DEMs for data filtering.

**Page 10, Lines 212 & 214:** "$R^2$"

**Page 11, Line 219:** "$R^2$"

**Figure 3 & Figure 4:** Why would RMSE in both figures be different? For example, for AW3D30 it is 13.4 m in Fig 3, but it is 34.9 m in Fig 4. Please make sure the data are consistent throughout the paper.

**Page 12, Line 240:** "The ME in the Himalaya is more negative than that in southeast Tibet", but it is not clearly shown in Figure 5. Do you want to quantify it with numbers?

**Page 12, Line 243:** Please be clear about "spatially relevant".

**Page 15, Line 278:** The time of six DEMs is all earlier than the ICESat-2 data. As stated in the major comments, in addition to uncertainties would temporal changes of the glaciers also affect your evaluation efforts?

**Figure 6:** No significant information is presented in Fig 6(d). You may just deleted it.

**Page 16, Line 309:** Replace "that is" with "they are".

**Page 16, Line 317:** Explain the numbers of "9, 3, 4, 3 and 1".

**Figure 10(a):** The figure displays the proportion of outliers vs. slopes. Within the range of 55-90 degrees, the proportion is up to ~40-55% for six DEMs. This is a high percentage. Would you please discuss the reasons?

**Page 21, Line 384:** "ICESat-2 elevation is … higher in Fig. 11d". This statement is not true in Fig.11d. Please describe the figure accurately and objectively.

**Page 22, Line 393:** Remove "will".

**Page 22-23, Lines 401-402:** Remove the sentence "Additionally, … elevation". The error of these two scatterings cannot influence the magnitude of the biased estimate of elevation you mentioned here.

**Page 26, Line 449:** Explain how is this correction of sub-pixel misregistration performed.

**Page 27, Line 470:** Change "A +5° error" to "An error of +5°".

**Page 28, Lines 495-497:** I cannot get what you mean here. Rewrite it to make it clear.

**Page 29, Lines 527-529:** It is not clear what you want to say here. It is in the conclusions section. You need to be brief and direct. Please rewrite.

---

## Community Comment (CC2)

[revised manuscript text omitted]

235 of excluded outliers relative to record of each DEM is excluded less than 1% of the data. Overall, there is non- irregular deviation existed among these DEMs and ICESat-2 elevation after filtering. The ICESat-2 vs DEMs values are distributed tightly around the fit line with a slope coefficient close to of 1, with no obvious differences among the $R^2$. NASADEM and TanDEM-X performed the best in terms of intercept and fit RMSE, with very little difference to the ICESat-2 data. For the other four DEMs, there are obvious systematic shifts which can be inferred from the high $R^2$ values, but high intercept values.

[Figure]

240

**Figure 3**. Differences between six DEMs and ICESat-2 elevation. a) AW3D30, b) SRTM-GL1, c) NASADEM, d) TanDEM-X, e) SRTM v4.1, and f) MERIT. The gradually lighter red lines denote the range within 2, 4 and 6 std of the mean. The text at the top left of each panel gives the fit results for data within 4 std of the mean. 'Outlier' denotes the proportion of outliers relative to the total records. 'R2', 'RMSE',

245  and 'Intercept' are fit results when the slope coefficient is set to 1. Elevation range was cut to 3500−6500 m, the range in which most elevations values are located, to show clearly the effect of using different multiples of the std from the mean.

The difference statistics for the six DEMs are presented in Figure 4. Statistically, Median and ME differed little, which indicated that  extreme values did not influence the ME much after the 4 std filter was applied. STD was slightly larger

250  than NMAD, especially for TanDEM-X, indicating larger discrepancies due to the DEM errors and noise (Höhle and Höhle 2009). NASADEM performed  better than the other two 30-m resolution DEMs in  ME . AW3D30 behaved best in RMSE (11.3 m), MAE (8.2 m). ~~has a lower absolute accuracy (RMSE: 34.9 m, ME: 32.3 m), but a similar relative accuracy to NASADEM because of the similar overall dispersion (~13 m) and spatial scale.have large differences~~differed

255  in RMSE (13.5 vs 12.6 m), MAE (10.0 vs 9.4 m), and ME (2.0 vs 0.9 m). The new algorithm and auxiliary data applied in NASADEM do indeed  improve the absolute accuracy of the product over glacierized terrain. The quality of TanDEM-X was the best out of the 90-m resolution DEMs with smallest RMSE (15.1 m), MAE (8.9 m), ME (−0.1 m), and STD (15.1 m). SRTM v4.1 and MERIT are both error-reduced products from SRTM3 v2 (Reuter et al. 2007; Yamazaki et al. 2017), and they have similar  ME 1.5 m vs  2.6 m) and RMSE (17.0 m. vs 15.6 m).

[Figure]

[Figure]

**Figure 4.** Overall difference (m) statistics between six DMEs and ICESat-2 elevation. a) 30-m-resolution DEMs, AW3D30, SRTM-GL1 and NASADEM. b) 90-m-resolution DEMs, TanDEM-X, SRTM v4.1 and MERIT. The vertical dash line denote the mean difference of each DEM between ICESat-2.

265

The spatial distribution of the ME and STD are shown in Figure 5. The ME  in southeast Tibet  is more  positive than that  in southeast Tibet the Himalaya, and it is slightly  negative in western Kunlun and the Karakoram mountains. It is worth noting that in the Himalaya and southeast Tibet, the ME of  these four DEMs is more  positive than that of TanDEM-X and AW3D30. ME of TanDEM-X are mainly at $\pm5$ m, but with some large values in several regions. SRTMGL-1, NASADEM, SRTM v4.1 and MERIT have nearly same distribution of ME, and all show negative ME values in the West Kunlun and Karakoram. ME of NASADEM is smaller than SRTM-GL1 in most regions of TP, but is bigger in West Kunlun and Karakoram. Overall, STD of 30-m resolution DEMs is much better than that of 90-m resolution DEMs (Fig.5b). STD along the Hindu Kush-Himalaya and southeast Tibet was larger than that in other regions. Thereinto, STD in southeast Tibet was relatively larger (>12 m). Specifically, the STD of AW3D30  was minimum and spatially relevant. Relative to ME, STD of NASADEM improved over the most part of TP, comparing 
[revised manuscript text omitted]

---

## Author Comment (AC1)

My detailed comments on the manuscript can be found in the attached annotated pdf. The manuscript discusses a relevant topic (accuracy of different DEMs on the Tibetan Plateau and impact on ice thickness reconstruction) and overall I find that the methods are robust and useful conclusions are drawn. I do have quite a few comments (see attachment), most prominently related to 1) the way differences are calculated [I suggest to subtract the ICESat-2 data from the DEM datasets rather than the other way around; this would avoid a lot of confusion], 2) the large number of figures [some suggestions for removing figures are given], 3) clarification or explanation [in many places details about the methods and presented figures and tables are missing; additionally, numerous textual revisions may help to clarify the content], 4) transferability of results to other regions [some discussion/speculation on how the presented findings on DEM and ice thickness comparison may translate to other mountainous regions would be interesting to add].

Response: Many thanks to the reviewer for the critical comments. We have carefully addressed all the comments. 1) And we have changed the way differences are calculated, Tables and Figures are all updated. 2) We have deleted two figures and simplified two figures. 3) Details about the method and figures are added. 4) Further discussions are also given in the manuscript. Our point-by-point responses are attached below in blue, while the original reviewers' comments are in black.

1.  "by ICESat-2" can be removed here
**Response:** Deleted

2. Line 16 Is this really considered?
**Response:** Yes, glacier elevation change data from Shean et al. (2020) is adopted in the evaluation, to distinguish the effect from glacier dynamics.

3. Why are the acquisition dates relevant in this sentence?
Response: Because glaciers elevation is changing all the time, if DEM across the large region is acquired in different periods, the estimated ice thickness by this DEM would not represent the real state of glacier. We revised this sentence.
"Considering the necessity of DEMs with consistency acquisition dates, NASADEM would be a best choice for ice-thickness estimates over the TP."
4. offered-> offers"
**Response:** Done

5. The effect of DEM grid resolution, giving more detailed slope information, on thickness estimation could be introduced here.
**Response:** We add one sentence here.
"Therefore, the DEM grid resolution could influence the thickness estimation from GlabTop, more detailed slope information could be provided by higher resolution DEM."

6. Slope or height?
**Response:** that's elevation error, we have corrected it.

7. Could be worth referring to Koldtoft et al. (2021) here.

**Response:** Done.

8. Could be good to shed some light on the relative significance of DEM errors compared to other uncertainties (particularly model physics).

**Response:** In this paragraph, we emphasized the fundamental role of DEM in determining the model physics, such as center flow lines, shear stress, apparent mass balance in line 42-44. Therefore, we add the sentence here as a conclusion after the literature review.

"DEM errors influence the determination of model physics and the final model outcomes."

9. Line 65 physiognomy?

**Response:** replaced with "landform"

**10.** Line 70 **something wrong with the grammar here.**

**Response:** We revised the sentence.

**"**publicly available DEM with high resolution**"**

11. I suppose the spatial coverage of ICESat-2 limits its use as a distributed DEM? It could be good to mention this, otherwise the reader may wonder why ICESat-2, rather than any of the 6 other DEMs, is not directly considered as the best DEM for thickness inversion.

**Response:** Yes, the reviewer is right, we revised this sentence.

"against ICESat-2 data which has been proven to have a high vertical accuracy and resolution but with sparse tracks (Fig. 1)."

12. That seems a bit superfluous after what is done in the previous assignment...

**Response:** Deleted.

13. Any indication of accuracy outside the Antarctic?

**Response:** We add one study in Qilianshan.

"The ATL06 product has better than 5 cm height accuracy and better than 20 cm surface measurement precision in the Antarctic and Qilianshan (Brunt et al. 2019; Zhang et al. 2020)."

*Zhang, Y., Pang, Y., Cui, D., Ma, Y., & Chen, L. (2020). Accuracy Assessment of the ICESat-2/ATL06 Product in the Qilian Mountains Based on CORS and UAV Data. IEEE Journal of Selected Topics in Applied Earth Observations and Remote Sensing, 14, 1558-1571.*

14. Accurancy ->Accuracy

Response: Corrected.

15. Why fill gaps? It makes the comparison less independent.

**Response:** The data was provided with gaps filled by JAXA officially. In this article, we try to directly know the quantity of DEM product acquiring from the official. We didn't fill any gaps in this article.

16. This is not a sentence. Furthermore, it would be better to split this into 4), 5) and 6)

**Response:** We revised this paragraph. And split this into 4), 5) and 6)

17. Line 171 Farinotti-> Farinotti

**Response:** Corrected.

18. Line 174 what value is chosen here?

**Response:** fsl=0.8, we have added this value.

19. Line 179 Not sure why this is relevant here.

**Response:** Deleted.

20. Line 181 From what experience? Please clarify.

**Response:** The parameter is determined from previous study; we have revised this sentence.
" which is determined from Huss and Farinotti et al. (2012)."

21. Line 188 This is inverting the shallow ice approximation, right? If so, that could be mentioned.

**Response:** Yes, the reviewer is right, we have added this information.
   "The Ice Thickness Inversion Based On Velocity (ITIBOV) model is inverting from the shallow ice approximation, it obtains the ice thickness by combining the surface velocity field with the Glen ice flow law (Gantayat et al. 2014; Glen 1955):"

22. Line 191 What year or period of velocity data are used?

**Response:** Mean velocity over 1985-2019 was used. We have added this information.
"We used the mean velocity over 1985-2019 from ITSLIVE dataset (Gardner 2019 )"

23. Line 195 So some other shared parameters are different?

**Response:** Yes, some parameters are unique for specific model. For example, the apparent mass balance parameter is only used in HF model.

24. Line 200 one thickness per point? How is this weighting determined?

**Response:** We add these sentences here to explain our method to determine the weight.
"First, the ensemble ice thickness was the sum of the four models with four weights w1, w2, w3, and w4, respectively. The sum of four weights equals to 1. 70% of the GPR result are adopted as calibration data. 30% of the GPR result are adopted as validation data. Then, the four weights iteratively changed to achieve the minimal mean absolute error between calibration data and model result. Finally, the MAE between ensemble ice thickness and validation data are calculated."

25. Why no R values?

**Response:** R is usually used to estimate the linear correlation between variables. Here we used NMAD comparing with ME to assess the influence from extreme errors as Höhle and Höhle (2009) and Gdulová et al. (2020). Also the $R^2$ in Figure3 shown little difference among DEMs. Therefore, we didn't use R in this study.

26. Line 205 It is very confusing that the elevation differences and the ME, MAE are all defined by taking H_ICESat-2 minus H_DEM. Naturally, I would expect the opposite (H_DEM - H_ICESat-2) and I would suggest to change this. That would change the sign of all the "differences" presented in Figs. 3-7, but it would make more sense since the DEMs are compared against the ICESat-2 data, not the other way around.

**Response:** We redo the analysis as Reviewer's suggestion. The Tables and Figures are all updated

27. These do not seem to be excluded yet in Fig. 3 and 4. Please mention.

**Response:** Fig.3 shown the differences of different filter ranges. Only the values in the range of 4std are used for further analysis. The fit result and overall difference statistics in Fig.3 and 4 are all based on filtered data.

28. Line 210 All outliers in one DEM or all of them?

**Response:** Ration of outliers relative to each DEM is less than 1%. And this sentence is revised.

"The four standard deviations (that is 4 std) was chosen to filter on the differences between ICESat-2 and DEMs to filter out extreme outliers. Ration of excluded outliers relative to record of each DEM is less than 1% ."

29 Line 211 = regular? What is meant?

**Response**: We revised this sentence.

"Overall, there is no irregular"

30 Line 215 The ME is a more direct indicator for bias / systematic shifts

**Response**: Yes, the review is right. The ME is a more direct indicator that we used it in the follow analysis. Here, we set the slope coefficient close to 1, so the intercept value is equal to the ME as shown in Figure 4.

31. Line 225 but this is not shown right?

**Response**: By comparing the ME and Median, we found that they are almost same, which indicated that the influence from extreme value are little after the application of 4 std filter.

32. Line 295. please clarify what it exactly implies that STD is substantially larger than NMAD for Tandem-X.

**Response:** This indicated that outliers and noise may exit in the TanDEM and we discuss this in the Sect 4.1 and Figure 12b. That indicate that obvious errors exit in the steep region in the TanDEM product.

"indicating larger discrepancies due to the DEM errors and noise"

33. Line 225.It could be worth mentioning that the mean error (ME) is something that can easily be corrected for by applying a bias correction. With that in mind, the STD (rather than RMSE) might be the best measure of performance? This still gives the NASADEM as the best performer, but also AW3D30 has a low STD.

**Response:** Yes, the reviewer is right. When calculating the ME, the positive and negative biases cancel each other, making the error smaller. We used STD not only to prevent this, but also to measure the variation or dispersion of the error. We add one sentence at the end of Section 2.4 "When calculating the ME, the positive and negative biases cancel each other, making the error smaller; Therefore, the STD together with ME could be a complementary indicator for assessment."

34. Line 235.It is somewhat striking that four of the DEMs have a ME that is between -31 and -33 m. Is there an obvious explanation for this large similar bias with the ICESAT-2 data? Based on Fig. 5 it seems that even the spatial distribution of the bias is very consistent between these four DEMs.

**Response**: We checked the references of six DEMs and found that their references are different. AW3D30, SRTM-GL1, SRTM v4.1 and MERIT are above EGM96 geoid. And ICESat-2, NASADEM and TanDEM is above WGS84 ellipsoid (Table 1). We have unified them to WGS84 ellipsoid. All figures and tables are updated.

35. Line 275.Is the effect that is visible in Fig. 6c), with most negative values at low elevations, the effect of more rapid glacier thinning at these elevations?

**Response:** Yes, apart from the elevation error in DEM, we thought that this is due to effect of rapid glacier thinning. The error in the low elevation region (Zone 1) is largely reduced after removing the effect of glacier elevation change (Table 3).

36. Line 280. Is there really a need to assign ablation zone, accumulation zone and transition zones here?

**Response:** Yes. We think it is necessary. The glacier elevation change is always changing and has different characteristics in ablation and accumulation zones. So, assessments in previous studies exclude the glacier terrain. The six DEMs in this study are acquired in different months and years. The glacier elevation change indeed influences the elevation differences between DEMs and ICESat-2. We divide the glacier terrain into 3 zones to estimate the effect of glacier dynamic.

37. Line 286.This seems counterintuitive to me. The six DEMs are all older than the ICESat-2 data, meaning that the ablation area in the six DEMs is likely to have higher elevations than in the ICESat-2 data (assuming the ablation areas have thinned). That would give a more positive difference in zone 1 and 2 than in zone 3 and 4, where less surface height changes over time are expected.

**Response:** The reviewer is right. Previously, we subtract the DEMs elevation from ICESat-2, so there is a more negative difference in zone 1 and 2 than in zone 3 and 4. In this version, we have subtracted ICESat-2 from the DEMs elevation as Reviewer's suggest. Now, there is a more positive difference in zone 1 and 2 than in zone 3 and 4.

38. Line 295. It would be good to reformulate this. What I think Fig. 7 mainly shows is that the observed shift in the difference from zone 1 to zone 4 is a sign that thinning between the time of collection of the six DEMs and the ICESat-2 data is most pronounced in zones 1 and 2. This could be mentioned here. To some extent this effect can already be seen in Fig. 6c as well if I am correct.

**Response:** Yes, the reviewer is right. We add one more sentence to reformulate it.

"The observed shift in the difference from zone 1 to zone 4 is a sign that thinning or accumulation between the time of collection of the six DEMs and the ICESat-2 data is most pronounced in zones

1 and 2."

38. Line 302 The performance of the ITIMs for the different DEMs will depend on the model parameters.

**Response**: Yes, model parameters would influence the model output. However, we adopt same model parameters for the specific one model, but using different DEMs. By this, we explored the influence of DEMs on the outcome of the ITIMs. And we found that even with the same parameters, the same model using different DEMs have different outcomes (Figure 8 and 9). The DEM indeed influence the performance of the ITIMs.

**39.Line 314 criterion->** criteria
**Response**: Corrected.

40. Line 326 A quick summary of how the weights for the different models are determined would be helpful.
**Response:** We add this in Section 2.3
"First, the ensemble ice thickness was the sum of the four models with four weights w1, w2, w3, and w4, respectively. The sum of four weights equals to 1. 70% of the GPR result are adopted as calibration data. 30% of the GPR result are adopted as validation data. Then, the four weights iteratively changed to achieve the minimal mean absolute error between calibration data and model result. Finally, the MAE between ensemble ice thickness and validation data are calculated."

41 Line 330 Why not as a final experiment make a composite bed estimate with only the best four ITIM-DEM combinations?
**Response**: Here, we try to estimate the effect of DEM on ITIMs. So only the ice thickness from four ITIMs using same DEM is ensembled.

42. Line 355 Fix column width
**Response:** Done.

43. Might be good to split into two subsections (4.1 and 4.2) focusing on glacier elevation change and terrain separately.
**Response**: Done.

44. Line 366 Please clarify "from the same original data,"
**Response**: We means that the NASADEM and SRTM-GL1 are both generated from NASA's Shuttle Radar Topography Mission. We have deleted this sentence in this version, because the large difference is due to the reference difference, not the vertical shift.

45.Line 381 This error correction method should have been introduced in the methods. The related results (Table 3) belong to the Results rather than the Discussion. Also, more details are needed. Have the DEMs been corrected or the ICESat-2 dataset?
**Response:** We add more details about this method in Section 2.4. Only the ICESat-2 Data is corrected by the glacier elevation change data. Table 3 is a direct proof of the effect of glacier

elevation change on the assessment of DEM. We thought it may be included in this Discussion.

"Glacier elevation changed a lot at -21－17m/yr over the TP during 2000-2018 (Shean et al. 2020). Therefore, the disparity of acquiring date between ICESat-2 and six DEM (Table 1) could introduce large error due to this glacier dynamic. TanDEM-X and AW3D30 are acquired in different months and years (Table 1), it's hard to analyse the impact of glacier dynamic on accuracy assessment. However, the other four DEMs are produced from NASA's Shuttle Radar Topography Mission during the 11-day mission in February 2000. We selected ICESat-2 data acquired in February 2019 and 2020. Then the glacier elevation dynamic magnitude during February 2000 and February 2019/2020 are subtracted from the selected ICESat-2 elevation based on the mean glacier elevation change data from Shean et al. (2020). By comparing the elevation from the four DEMs and adjust ICESat-2, we could exactly know the impact from glacier dynamic."

46. Line 389 The figure is confusing. What is adjusted ICESat-2? It would make more sense to adjust the six DEMs and compare with the original ICESat-2 only. With DEMs collected in different years it is currently not clear for what period the ICESat-2 data are adjusted. The comparison currently does not make sense to me.

**Response:** Because the collect year of NASADEM, SRTM GL1 , SRTM v4.1 and MERIT is on Feb. 2000. Here we adjust the one track ICESat-2 data on the glacier to the year of 2000 using the glacier elevation change data over 2000-2018 from Shean et al. (2020). By this figure, we want to conclude that the difference between DEMs and ICESat-2 would be reduced after adjusting ICESat-2 elevation. However, in this version, we removed this figure as reviewer's suggest.

47. Line 399 Not sure what you want to say here. I suppose the ICESat-2 dataset gives average elevation over the 2018-2020 period, so no seasonal dependence. Am I right that the other DEMs reflect one point in time, e.g. Feb 2000. That can of course give some bias. It could be good to rephrase a bit here.

**Response**: The reviewer is right. We rephrased this sentence.

"When taking all points from different seasons into consideration, ICESat-2 dataset gives average elevation over the 2018-2020 period, the seasonal effects could also partly cancel each other out."

48. Line 403 Any rough idea on magnitude of this error?

**Response**: Corrections of these two errors require information about cloud structure and ice-surface conditions that are not available when ATL06 is processed. It remains an active avenue of research.

49 Line 409 Why show this? It is hard to see any differences between the panels this way.

**Response**: By this we try to show that there are serious errors in the steep region in TanDEM in ROI A region denoted in this figure. Of course, difference among the other five DEMs show little difference.

50. Line 416 The main thing I see in Fig. 13 is that there are many more measurements with north aspect, so both for steep and gentle slopes.

**Response**: If we fixed slope axis, we can find that there more measurements in north aspect at the same slope. That's to say that, there more measurements with steep slopes in the north aspect.

51. Line 433 Could be worth highlighting which DEM suffers most/least from slope effects. Fig 10 gives an indication but only focuses on extreme outliers rather than mean differences. I am also curious how accurate ICESat-2 is in steep terrain. I can only find estimates for (flat) Antarctica in this study.

Response: We add one sentence in Setction 3.2.

"Overall, relative to the other DEMs, AW3D30 and NASADEM behaves best against slope in terms of spread and median value."

A study in Qilian Shan in north TP shown a less than 20 cm accuracy. We have added this reference.

52. Line 433 It would be good to explain briefly what is meant with misregistration.

Response: We added one sentence to explain "misregistration"

"Pixel of different DEMs at same location may mismatch each other."

53.Line 439 I am currently not sure how to interpret Fig. 14. What are "offset pixels"? Is it a measure of how many pixels (of 30 or 90 m) a DEM is shifted relative to ICESat-2 within a 1 by 1 degree grid cell? Please explain.

Response: Yes, the reviewer is right. We estimate the offset distance of DEM by 1×1 degree relative to ICESat-2. Then this offset distance is converted to offset pixels according to the resolution of a DEM. After unifying the reference, the shift pixels are all within one pixel, and show little spatial difference. We have updated Figure 14.

54. remove "except for" since these seem to be the DEMs that are affected. (?)

Response: We have deleted this paragraph after updating the data.

55. I can't follow this reasoning. Please clarify.

Response: We have deleted this paragraph after updating the data. The shift pixels are all within one pixel and has little influence on the assessment.

56. Line 459 This is not shown in Fig. 16. Maybe in Fig. 14, but only in red?

Response: We have deleted this Figure after updating the data. The shift pixels are all within one pixel and has little influence on the assessment.

57. I find this section rather chaotic and parts of it are of limited value. The discussion of Fig. 9 is relevant and should be kept, but the discussion of slope and elevation range perturbations in the first paragraph and Fig. 16 do not add much new insight and in my opinion can be removed. The experiments are very hypothethical as a homogeneous slope perturbation and a elevation range perturbation are not something that one can expect to happen for real DEMs. From the methods in Section 2 it can already be concluded which models would be most sensitive to certain types of terrain errors.

Response: We discussed the reviewer's suggestions. Yes, the reviewer is right, we can conclude that which model would be sensitive to certain types of terrain errors from the methods in Section 2. However, we could not conclude the degree of sensitivity of different models to terrain factors. For example, we know that GlabTop2 would be sensitive to elevation and slope, we found that small

size glaciers would be more sensitive than big size glacier in our test. In the formula of HF and OGGM in the method, we may guess that these two models would be sensitive to the slope, but actually they have a good robustness to the accuracy of input DEM. We have deleted some irrelevant portions and readjusted the paragraph structure to solve reviewer's comments.

58 Line 464 where is this shown? It is not in Fig. 10 it seems.
**Response**:It's shown in Figure 8.

52 Line 465 and 45 % of what?
**Response:We rephased this sentence.**
"Generally, the outcome with GlabTop2 and ITBOV using 30-m DEMs is 51% and 43% better than with the 90-m DEMs in mean error, respectively."

53.Is this really needed? I would suggest to remove this part.
**Response**:Removed.

54.Line 505 please clarify or reformulate.
**Response**:We reformulate it as follow.
"When the results from different models are ensembled, "

55.Line510 I am missing discussion on how the results found in this study can potentially be of use for others using these DEMs, potentially as input in an ice thickness estimation model, in other mountainous (glacierized) regions than the Tibetan Plateau. Is it likely that the same conclusions could be drawn in other regions? Why (not)?
**Response**:We add further discussion here.

[revised manuscript text omitted]

235    of excluded outliers relative to record of each DEM is excluded less than 1% of the data. Overall, there is non- irregular deviation existed among these DEMs and ICESat-2 elevation after filtering. The ICESat-2 vs DEMs values are distributed tightly around the fit line with a slope coefficient close to of 1, with no obvious differences among the $R^2$. NASADEM and TanDEM-X performed the best in terms of intercept and fit RMSE, with very little difference to the ICESat-2 data. For the other four DEMs, there are obvious systematic shifts which can be inferred from the high R² values, but high intercept values.

[Figure]

240

**Figure 3**. Differences between six DEMs and ICESat-2 elevation. a) AW3D30, b) SRTM-GL1, c) NASADEM, d) TanDEM-X, e) SRTM v4.1, and f) MERIT. The gradually lighter red lines denote the range within 2, 4 and 6 std of the mean. The text at the top left of each panel gives the fit results for data within 4 std of the mean. 'Outlier' denotes the proportion of outliers relative to the total records. 'R2', 'RMSE',

245 and 'Intercept' are fit results when the slope coefficient is set to 1. Elevation range was cut to 3500−6500 m, the range in which most elevations values are located, to show clearly the effect of using different multiples of the std from the mean.

The difference statistics for the six DEMs are presented in Figure 4. Statistically, Median and ME differed little, which indicated that  extreme values did not influence the ME much after the 4 std filter was applied. STD was slightly larger

250 than NMAD, especially for TanDEM-X, indicating larger discrepancies due to the DEM errors and noise (Höhle and Höhle 2009). NASADEM performed  better than the other two 30-m resolution DEMs in  ME . AW3D30 behaved best in RMSE (11.3 m), MAE (8.2 m). ~~has a lower absolute accuracy (RMSE: 34.9 m, ME: -32.3 m), but a similar relative accuracy to NASADEM because of the similar overall dispersion (~13 m) and spatial scale.have large differences~~differed

255 in RMSE (13.5 vs 12.6 m), MAE (10.0 vs 9.4 m), and ME (2.0 vs 0.9 m). The new algorithm and auxiliary data applied in NASADEM do indeed  improve the absolute accuracy of the product over glacierized terrain. The quality of TanDEM-X was the best out of the 90-m resolution DEMs with small RMSE (15.1 m), MAE (8.9 m), ME (-0.1 m), and STD (15.1 m). SRTM v4.1 and MERIT are both error-reduced products from SRTM3 v2 (Reuter et al. 2007; Yamazaki et al. 2017), and they have similar  ME 1.5 m vs 2.6 m) and RMSE (17.0 m vs 15.6 m).

[Figure]

[Figure]

**Figure 4.** Overall difference (m) statistics between six DMEs and ICESat-2 elevation. a) 30-m-resolution DEMs, AW3D30, SRTM-GL1 and NASADEM. b) 90-m-resolution DEMs, TanDEM-X, SRTM v4.1 and MERIT. The vertical dash line denote the mean difference of each DEM between ICESat-2.

265

    The spatial distribution of the ME and STD are shown in Figure 5. The ME  in southeast Tibet  is more  positive than that  in southeast Tibetthe Himalaya, and it is slightly  negative in western Kunlun and the Karakoram mountains. It is worth noting that in the

270   Himalaya and southeast Tibet, the ME of  these four DEMs is more  positive than that of TanDEM-X and AW3D30. ME of TanDEM-X are mainly at $\pm 5$ m, but with some large values in several regions. SRTMGL-1, NASADEM, SRTM v4.1 and MERIT have nearly same distribution of ME, and all show negative ME values in the West Kunlun and Karakoram. ME of NASADEM is smaller than SRTM-GL1 in most regions of TP, but is bigger in West Kunlun and Karakoram. Overall, STD of 30-m resolution DEMs is much better than that of 90-m resolution DEMs (Fig.5b). STD along

275   the Hindu Kush-Himalaya and southeast Tibet was larger than that in other regions. Thereinto, STD in southeast Tibet was relatively larger (>12 m). Specifically, the STD of AW3D30  was minimum and spatially relevant. Relative to ME, STD of NASADEM improved over the most part of TP, comparing 
[revised manuscript text omitted]

---

## Author Comment (AC2)

Comments on

"Towards ice thickness inversion: an evaluation of global DEMs by ICESat-2 in the glacierized Tibetan Plateau" by Wenfeng Chen et al.

Referee #2

Overview

This paper by Chen et al. presents a method to evaluate existing regional scale DEMs using the recently available ICESat-2 elevation product. The DEMs are then applied to model the ice thickness of glaciers in the Tibetan Plateau region. The quality of the inversion results is then analyzed to prove the effectiveness of the ICESat-2 based evaluation. The paper is generally well structured. It showed that the ICESat-2 data provided a comparison dataset for selecting an optimal DEM that can be used as input for ice thickness inversion. This work should be useful to researchers of The Cryosphere community. The paper needs to be revised according to the following major and minor comments.

The manuscript needs a serious improvement of both formal English writing and scientific meaning.

**Response**: We are grateful to the anonymous reviewer for the constructive comments on our manuscript. We have carefully addressed all the comments below. The English are checked by a native English speaker. We have added some content to improve the scientific meaning.

   "This conclusion is of significance for ice thickness inversion models using DEMs in TP. However, it should be noted that the result may be not suitable for studies in other glacierized mountainous regions. Because various errors exist in DEMs, such as speckle noise, stripe noise and absolute bias; they behave differently across the Earth (Yamazaki et al. 2017; Takaku et al. 2020). But our method to assess the accuracy of DEMs is repeatable in different regions, combined with the recently released glacier elevation change data on Earth (Hugonnet et al. 2021). What's more, benefiting from the high accuracy and dense coverage of ICESat-2 data, the quality of DEMs can also be improved as similar as the production of MERIT (Yamazaki et al. 2017). For example, the misregistration in DEMs could be corrected and terrain-related errors could be reduced by unitizing the relation of difference against slope, aspect and elevation in Fig. 6."
"

Major comments

1. The subtitles in the Data and Methods sections need to be improved to reflect the actual contents and be logic (ICESat-2 elevation data referenced, DEMs evaluated, ……)

**Response**: The subtitles are revised and organized to reflect the actual contents.

2.1 Descriptions of ICESat-2 elevation data referenced;

2.2 Descriptions of global-scale DEMs evaluated;

2.3 Ice thickness inversion method;

2.4 Accuracy assessment method;

2. The ICESat-2 data were used to evaluate DEMs. It should be discussed how the
DEMs were generated, including data sources, time periods, and uncertainties.
During the time differences between the ICESat-2 data and MEDs there may be
glacier surface changes that may affect the evaluation. If this is not considered,
would this also affect the thickness inversion results?

**Response**: We add more details about how the six DEMs are generated. Detailed information is also summarized in Table1.

Yes, glacier surface change affects the evaluation. We considered it in the analysis in Section 3.3 and discussed it in section 4.1. We also added one paragraph in Section 2.4 to describe how we solve the influence from glacier surface change on evaluation.

"Glacier surface elevation changed at $-21-17$m/yr over the TP during 2000-2018 (Shean et al. 2020). Therefore, the disparity of acquiring date between ICESat-2 and six DEM (Table 1) could introduce large error due to the glacier dynamic. TanDEM-X and AW3D30 are acquired in different months and years (Table 1), it's hard to analyse the impact of glacier dynamic on accuracy assessment. However, the other four DEMs are produced from NASA's Shuttle Radar Topography Mission during the 11-day mission in February 2000. We selected ICESat-2 data acquired in February 2019 and 2020. Then the glacier elevation dynamic magnitude during February 2000 and February 2019/2020 are subtracted from the selected ICESat-2 elevation based on the mean glacier elevation change data from Shean et al. (2020). By comparing the elevation from the four DEMs and adjust ICESat-2, we could exactly know the impacts on accuracy assessment from glacier dynamic."

Glacier surface elevation change could also influence the inversion of ice thickness, especially when estimating glacier thickness in regional scale. Therefore, though AW3D30 with mixing acquiring dates exhibit a good accuracy assessment, we still suggest NASADEM is a best choice for ice-thickness estimates over the TP.

3. ICESat-2 Level-3A land-ice ATL06 product was used in this work. There should be
a good understanding of the quality of this product itself, although a systematic calval may not have been
performed in the glacierized Tibetan Plateau region. I would
like to see even a general discussion in that regard. I suggest to add some relevant
references in the Data section:
Brunt, K. M., Neumann, T. A., and Smith, B. E.: Assessment of ICESat-2 Ice Sheet
Surface Heights, Based on Comparisons Over the Interior of the Antarctic Ice Sheet,
Geophys. Res. Lett., 46, 13072–13078, https://doi.org/10.1029/2019GL084886,
2019.
Brunt, K. M., Smith, B. E., Sutterley, T. C., Kurtz, N. T., and Neumann, T. A.:
Comparisons of Satellite and Airborne Altimetry With Ground-Based Data From
the Interior of the Antarctic Ice Sheet, Geophys Res Lett, 48, e2020GL090572
https://doi.org/10.1029/2020GL090572, 2021.
Li, R., Li, H., Hao, T., Qiao, G., Cui, H., He, Y., Hai, G., Xie, H., Cheng, Y., and
Li, B.: Assessment of ICESat-2 ice surface elevations over the Chinese Antarctic
Research Expedition (CHINARE) route, East Antarctica, based on coordinated
multi-sensor observations, The Cryosphere, 15, 3083–3099,
https://doi.org/10.5194/tc-15-3083-2021, 2021.

**Response**: We have added the above reference.

"The segment has a length of 40 m centered on reference points at 20-m intervals along the track. The ATL06 product has better than 5 cm height accuracy and better than 20 cm surface measurement precision in the Antarctic (Brunt et al. 2019,2020; Li et al. 2021) and Qilian Shan (Zhang et al. 2020)."

Elevation differences between crossovers formed by ICESat-2 tracks may be used
for the elevation accuracy evaluation.

**Response**: We thanks for the reviewer's suggestion. This is really a good idea for testing the performance of ICESat-2 in stable region. However, when we check the dates of point around crossover of ICESat-2 tracks, we find they are from different dates, so maybe it's not easy to know whether this elevation difference is from glacier elevation change or the uncertainty of ICESat-2.

As ATL06 product comes from ATL03 photons grouped in 40 m segments, would
ATL03 data provide more terrain details? This may be outside of the scope of this
paper. However, a discussion of this potential would be helpful to the readers in
their future research.

**Response**: We added these sentences in Section 2.1.

"ICESat-2 ATL03 and ATL06 product both can be used as elevation reference. ATL03 product has a spacing of ~0.7m and can provide more terrain details than ATL06 product. In this study, considering the resolution of global dem and compute cost, we select the ICESat-2 Level-3A land-ice ATL06 product as elevation reference."

4. Based on your results, the accuracy of ICESat-2 data is better than the compared
global-scale DEMs. Have you considered to use the ICESat-2 data to improve the
quality of the DEMs, especially in areas where ICESat-2 along and across track data
are available. Again, this may be considered as a future work

**Response**: Yes, that's a good point. We added this in the discussion.

"What's more, benefiting from the high accuracy and dense coverage of ICESat-2 data, quality of DEMs can also be improved as similar as the production of MERIT (Yamazaki et al. 2017). For examples, the misregistration in DEMs could be corrected and terrain-related errors could be reduced by unitizing the relation of difference against slope, aspect and elevation in Fig. 6."

**Minor comments**

**Page 1, Line 11:** Replace "derived" with "derived from".

**Response**: Corrected

**Page 1, Lines 12–14:** This sentence is awkwardly phrased. You may rephrase by separating it in to two sentences.

**Response**: Corrected

"However, the scarce in-situ measurements of glacier surface elevation limit the evaluation of DEM uncertainty. And hence influence of DEM uncertainty on ice-thickness modelling remains unclear over the glacierized area of the Tibetan Plateau (TP)."

**Page 1, Line 17:** Please be clear if it is the horizontal or vertical accuracy.

**Response**: it's vertical accuracy. Corrected

**Page 1, Lines 23-24:** Change "one pixel" to one grid spacing.

**Response**: Corrected.

**Page 1, Lines 24-28:** These sentences are not communicating well. Please cut them to short and simple sentences.

**Response**: Corrected

Then, influence of six DEMs on four ice-thickness models: GlabTop2, Open Global Glacier Model (OGGM), Huss-Farinotti (HF), Ice Thickness Inversion Based on Velocity (ITIBOV) was intercompared. The results show that GlabTop2 is sensitive to the accuracy of both elevation and slope, while OGGM and HF are less sensitive to DEM quality and resolution, and ITIBOV is the most sensitive to slope accuracy.

Page 2, Lines 33 & 36: Change "km2 ", "km3 " to "km2", "km3".

**Response**: Corrected

Page 3, Line 87 and Page 4, Lines 107-108: When you mention ICESat-2 in these two places, please mention the cal-val efforts and introduce the most recent accuracy assessment results by Brunt et al. (2019, 2021) and Li et al. (2021).

**Response**: Corrected.

"MERIT which are derived from different sensors and have different resolutions, against ICESat-2 data which has been proven to have a high vertical accuracy and resolution (Brunt et al. 2019, 2021; Li et al. 2021)"

"The ATL06 product has better than 5 cm height accuracy and better than 20 cm surface measurement precision in the Antarctic (Brunt et al. 2019,2020; Li et al. 2021) and Qilian Shan (Zhang et al. 2020)."

Page 4, Line 95: Replace "intersecting" with "covering".

**Response**: Corrected

Page 4, Line 105: ~17 m diameter is the design value. This value needs to be updated according to the new study: Magruder, L. A., Brunt, K. M., and Alonzo, M.: Early ICESat-2 on-orbit geolocation validation using ground-based corner cube retroreflectors, Remote Sensing, 12, 3653, https://doi.org/10.3390/rs12213653, 2020.

**Response**: Corrected.

"ICESat-2 ATLAS (Advanced Topographic Laser Altimeter System) emits a pulse every 0.7 m along the track covering a horizontal circular area with 0.5 m in vertical extent and ~17 m diameter. This design diameter value varied due to the photo-counting lidar technology and potentially the atmospheric conditions (Magruder et al. 2020)."

**Page 4, Line 106:** In addition to first-photon bias, transmit-pulse shape correction should be mentioned also

**Response**: Corrected.

"We used the ICESat-2 Level-3A land-ice ATL06 product. ATL06 heights are median-based heights derived from a linear-fit model over each segment corrected for first-photon bias and transmit-pulse shape."

The ATL06 data provide slopes in along and across tracks. They are derived at different scales (along track with denser points and cross track with fewer points and longer separations). Have you considered the difference between these two

types of slopes?

**Response**: Here, the slope is derived from DEMs not the ICESat-2 product. We use this slope for error analysis at slope scale and misregistration analysis. In the origin design of this research, we plan to estimate the slope accuracy derived from DEMs based on the ICESat-2 along-track slope. But we found that the algorithms they calculated slope are totally different (Burrough and McDonell 1998; Smith et al. 2019)., so we didn't estimate this furtherly.

Burrough, P.A., & McDonell, R.A. (1998). Principles of Geographical Information Systems. *Oxford University Press, New York*, 190 pp.

Smith, B., Fricker, H.A., Holschuh, N., Gardner, A.S., Adusumilli, S., Brunt, K.M., Csatho, B., Harbeck, K., Huth, A., Neumann, T., Nilsson, J., & Siegfried, M.R. (2019). Land ice height-retrieval algorithm for NASA's ICESat-2 photon-counting laser altimeter. *Remote Sensing of Environment, 233*

*Figure 2: "Difference in aspect and slope bins".*

**Response**: Corrected.

*Figure 2: Make different blue boxes into just blue color.*

**Response**: Corrected.

[Figure]

*Page 6, Line 127: Please specify how severe is the gap situation and the effectiveness of gap filling*

**Response**: We have added these sentences.

"Approximate 10 % of global land area, mainly in tropical rainforest areas and the polar areas, has voids mostly due to cloud or snow/ice covers constatation in source imageries. Data gaps are filled with SRTM, ASTER GDEM v3, ArcticDEM v3, and TanDEM-X 90 (Takaku et al. 2020). After filling gaps, the accuracies in void-filled and void-free areas are nearly consistent (Takaku et al. 2020)."

*Page 6, Line 135: All website links need to add the last visit date to ensure their availability*

**Response**: We have added visit date of website links in Table 1.

**Page 8, Lines 160-162:** The sentence is unclear. Please rewrite.

**Response**: corrected.

*"Four ice-thickness inversion models (GlabTop2, HF, OGGM, ITBOV) were used to estimate the glacier thickness. The Chhota Shigri Glacier located in western Himalaya with available GPR data (Fig.1) was selected as an example to evaluate the influence of DEM uncertainty on the ITIMs."*

**Page 10, Line 209:** Please justify your choice of 4-std of differences between ICESat-2 and DEMs for data filtering.

**Response**: We add these sentences.

*"The four standard deviations (that is 4 std) was chosen to not only filter the differences between ICESat-2 and DEMs to exclude extreme outliers, but also keep most records in the further accuracy analysis. Ration of excluded outliers relative to record of each DEM is less than 1%."*

**Page 10, Lines 212 & 214:** "R2"

**Response**: Corrected.

**Page 11, Line 219:** "R2"

**Response**: Corrected.

**Figure 3 & Figure 4:** Why would RMSE in both figures be different? For example, for AW3D30 it is 13.4 m in Fig 3, but it is 34.9 m in Fig 4. Please make sure the data are consistent throughout the paper.

**Response**: The RMSE in Figure 3 and Figure 4 are different. 'R2', 'RMSE' and 'Intercept' are fit results when the slope coefficient is set to 1. In fact, the RMSE in Figure 3 should be equal to the STD in Figure 4. The intercept should be equal to the ME in Figure 4.

**"**The ME in the Himalaya is more negative than that in southeast Tibet", but it is not clearly shown in Figure 5. Do you want to quantify it with numbers?

**Response:** The Figure 5 is updated. And the sentence is rewritten.

"The ME in southeast Tibet is more positive than that in the Himalaya"

[Figure]

**Page 12, Line 243:** Please be clear about "spatially relevant"

**Response:** Corrected.

" Specifically, the STD of AW3D30 and NASADEM was minimum and has similar spatial distribution."

Page 15, Line 278: The time of six DEMs is all earlier than the ICESat-2 data. As stated in the major comments, in addition to uncertainties would temporal changes of the glaciers also affect your evaluation efforts?

**Response:** Yes, the glacier elevation change could also influence the evaluation. We quantitatively estimate the influence from glacier elevation change in Section 3.3 and 4.1. Especially for glaciers in the ablation zone, the ME, MAE SD and RMSE are largely reduced for SRTM based DEMs (Table 3).

**Figure 6:** No significant information is presented in Fig 6(d). You may just deleted it.

**Response:** We used this Fig.6d here.

"For NASADEM and SRTM-GL1, the differences along the elevations show similar distribution and varied from –10 to 10 m over the range 4500−6500 m, where measurements are concentrated (Fig. 6d);"

"The differences with aspect show contrasting features to the distribution of measurements in different aspects (Fig. 6d)."

"Though points in 55º−90º slope region account for small fraction (Fig. 6d), almost half the points in the 55º−90º slope region are identified as extreme outliers (Fig. 10a)."

**Page 16, Line 317:** Explain the numbers of "9, 3, 4, 3 and 1".

**Response:** We refined this sentence and title of Table 2.

"Totals of 8, 7, 3 and 2 output achieved the minimum RMSE in profiles (Bold number in Table 2) by different ITIMs using AW3D30, NASADEM, TanDEM-X and SRTM-GL1, respectively."

"Table 2 RMSE (m) of modelled ice thickness compared with ground penetrating radar (GPR)

measurements on each profile. Location of profiles are shown in Figure 1. Bold numbers denote the best model performance on each profile using different DEMs."

The figure displays the proportion of outliers vs. slopes. Within the range of 55-90 degrees, the proportion is up to ~40-55% for six DEMs. This is a high percentage. Would you please discuss the reasons?
**Response:** We discussed in the section 4.2. We thought that the steep slope and intra-pixel effect should be attributed to this.

 "Almost half the points in the 55°−90° slope region are identified as extreme outliers (Fig. 10a). Differences also show large discrepancies for all DEMs in the steeply sloping regions where voids and large errors are frequent (Falorni 2005). Steep slopes combined with low resolution led to variations in the spread of differences in Fig. 6b. Spreads of differences were larger on steep slopes for the 90-m DEMs than those of the 30-m DEMs. Intra-pixel variation aggravates this effect in steeply sloping regions (Uuemaa et al. 2020), lower resolution or reduced pixel DEMs smooth the terrain details and lead to inaccurate elevation compared with the 20-m footprint of ICESat-2 points. The spread and the number of outliers gradually increased with the slope, especially for the TanDEM-X case (Fig. 7b)."

**Page 21, Line 384:** "ICESat-2 elevation is … higher in Fig. 11d". This statement is not true in Fig.11d. Please describe the figure accurately and objectively
**Response:** Since the profile overlap seriously, we deleted this Figure 11.
**Page 22, Line 393:** Remove "will".
**Response:** Corrected.
**Page 22-23, Lines 401-402:** Remove the sentence "Additionally, … elevation". The error of these two scatterings cannot influence the magnitude of the biased estimate of elevation you mentioned here.
**Response:** Corrected.

**Page 26, Line 449:** Explain how is this correction of sub-pixel misregistration performed.
**Response:** Corrected.

"According to the sinusoidal relationship between aspect and error differences between two DEMs (Van Niel et al. 2008), using the co-registration method in Nuth and Kääb (2011) and ICESat-2 points outside the glaciers, offset pixels relative to ICESat-2 in x- and y- direction at the 1°×1° grid scale were estimated by fitting method in MATLAB across the TP."

**Page 27, Line 470:** Change "A +5° error" to "An error of +5°".
**Response:** Corrected.
**Page 28, Lines 495-497:** I cannot get what you mean here. Rewrite it to make it clear.
**Response:** It's repeated in this paragraph. So, it's deleted.

**Page 29, Lines 527-529:** It is not clear what you want to say here. It is in the conclusions section. You need to be brief and direct. Please rewrite.
**Response:** Corrected. Some sentences are also deleted to make the conclusion section brief.

[revised manuscript text omitted]

---

## Referee Report (RR1)

**Comments on "Towards ice thickness inversion: an evaluation of global DEMs in the glacierized Tibetan Plateau" by Wenfeng Chen et al.**

**General Comments**

The authors have made good efforts to address my comments on the previous version of the manuscript. They answered all my questions. It should be noted that my review and comments mainly focus on evaluation of DEMs using ICESat-2 and influence of DEMs on the inversion quality. The evaluation of the influence of the various inversion models on the thickness is out of my expertise.

One point that still needs to be addressed is introduction of the ground penetrating radar (GPR) data used for validation of the inversed thickness, in terms of equipment, location distribution, acquisition time, and quality. It may be a short subsection in the data section. There should be a clarification later as how they are used for validation.

After addressing the GPR point and following minor edits, the manuscript is good for publication from my point of view.

**Specific comments:**

There are some minor edits in the attached PDF.

[revised manuscript text omitted]

---

## Referee Report (RR2)

[referee-annotated manuscript omitted]

---

## Author Response (AR2)

Dear Dr. Wenfeng Chen,

Thank you for the revised version. You will see that both reviewers recommend minor revision and I generally concur with their assessments. Please pay careful attention to their remarks (see also appended manuscripts), as well as my own comments in another appended manuscript (mostly pertaining to your figures). Please detail in an authors's response the comments by the reviewers and myself, and the actions taken by you and your colleagues.

A final recommendation of acceptance will perhaps be facilitated by another editor as I will shortly leave for Antarctica.

Best wishes,

Arjen Stroeven

Dear Editor Arjen Stroeven,

Thanks for your prompt and professional process of our manuscripts. We have carefully addressed all the comments from you and two reviewers. Our point-by-point responses are attached below in blue, while the original reviewers' comments are in black. Wish you a nice experience in Antarctica!

Sincerely,
Wenfeng Chen

**Comments from Editor Arjen Stroeven**
Line 97: "Covering with" -> "and"
Corrected.

Line 98: (for location, see a)
Added.

Line 101: Here, and elsewhere, "m a.s.l."
Corrected.

Line 123: ITBOV or ITIBOV?
It's ITIBOV, we have corrected throughout the manuscript.

Line 165: "." Remove
Removed.

Line 165: SRTM v4.1
Corrected.

Line 165: direct to an open database instead?

Yes, this link is direct to an open database, the data is shared by Google Drive.

Line 212: define MAE
It's mean absolute error, and we have defined it.

Line 240: sigma? Refer to panel d
It's sigma, we have corrected it.

Line 255: explain abbreviations used in the caption: RMSE etc..
We have added the definition of abbreviations in the caption.
"In bottom panel, ME is Mean error, MAE is mean absolute error, Median is median error, RMSE is root mean square error, STD is standard deviation, and NMAD is normalized median absolute deviation."

Line 255: "dash line" -> "dashed lines"
Corrected.

Line 273: Standard deviation (m)
Corrected.

Line 275: "."
Added

Line 297: these values don't seem to fit aspect and slope in (d)? Aspect appears to be binned by 10 degrees and slope I'm unsure but >2?
Figure 6d is updated. Now the bin gap is consistent with Figure 6a−c.

[Figure]

Line 321: that-> as
Corrected.

Line 343: please make sure that all scales have the same range (so that the same color in all panels mean the same ice thickness. Hard to compare otherwise.
We have improved this figure, using the same scale range for all subplots.

[Figure]

a) GlabTop2 AW3D30        SRTM-GL1        NASADEM        TanDEM-X        SRTM v4.1        MERIT

ME (m)   67.1              65.4             87.8             37.5             37.0             39.3
MAX (m)  214.4             182.4            199.5            140.5            138.0            173.4

b) HF

ME (m)   84.4              82.4             87.8             82.7             85.0             84.1
MAX (m)  396.4             318.2            365.8            345.6            371.9            393.1

c) ITIBOV

ME (m)   86.7              84.0             87.1             58.6             61.8             64.6
MAX (m)  262.7             269.8            257.0            238.8            250.3            241.7

d) OGGM

ME (m)   98.0              97.1             82.4             93.1             90.0             96.7
MAX (m)  331.9             360.2            318.2            331.0            303.9            353.0

e) Composite Model

ME (m)   98.0              93.9             93.1             89.7             95.5             96.5
MAX (m)  312.3             310.1            353.3            329.7            317.8            333.7

Thickness (m)                                              1 km

0   40   80   120  160  200  240  280  320  360  400

Line 356: ITBOV in figure

Corrected.

Line 365: spell out RMSE

Added.

Line 383: would it not make more sense to label them C and D to refer to panels c and d?

Figure 10 is improved.

[Figure]

Line 406: m a.s.l.
Corrected.

Line 455: "Figure 8 and 9" -> "Figures. 8, 9"
Corrected.

Line 575: "make" -> "made", "estimate" -> "estimated"
Corrected.

**Reviewer #1**

Comments on "Towards ice thickness inversion: an evaluation of global DEMs in the glacierized Tibetan Plateau" by Wenfeng Chen et al.

General Comments

The authors have made good efforts to address my comments on the previous version of the manuscript. They answered all my questions. It should be noted that my review and comments mainly focus on evaluation of DEMs using ICESat-2 and influence of DEMs on the inversion quality. The evaluation of the influence of the various inversion models on the thickness is out of my expertise.

One point that still needs to be addressed is introduction of the ground penetrating radar (GPR) data used for validation of the inversed thickness, in terms of equipment, location distribution, acquisition time, and quality. It may be a short subsection in the data section. There should be a clarification later as how they are used for validation. After addressing the GPR point and following minor edits, the manuscript is good for publication from my point of view.

We thank the reviewer for the valuable comments and suggestions, which improved our manuscript a lot. The GPR data used in this study is acquired from Azam et al. (2017), and this data is open

access at Farinotti et al. (2021). Detailed information about the GPR data is introduced by Azam et al. (2017). We have added one sentence in Section 2.3.

*"The GPR data were measured based on a pulse radar system in October 2009 (Azam et al., 2017) and is available at Farinotti et al. (2021)."*

Reference:

Azam, M.F., Wagnon, P., Ramanathan, A., Vincent, C., Sharma, P., Arnaud, Y., Linda, A., Pottakkal, J.G., Chevallier, P., Singh, V.B., & Berthier, E: From balance to imbalance: a shift in the dynamic behaviour of Chhota Shigri glacier, western Himalaya, India. J GLACIOL, 58, 315-324, https://doi.org/10.3189/2012JoG11J123,2012

Farinotti, D., Brinkerhoff, D.J., Fürst, J.J., Gantayat, P., Gillet-Chaulet, F., Huss, M., Leclercq, P.W., Maurer, H., Morlighem, M., Pandit, A., Rabatel, A., Ramsankaran, R., Reerink, T.J., Robo, E., Rouges, E., Tamre, E., van Pelt, W.J.J., Werder, M.A., Azam, M.F., Li, H., & Andreassen, L.M.: Results from the Ice Thickness Models Intercomparison eXperiment Phase 2 (ITMIX2). Front. Earth. Sci., 8, https://doi: 10.3389/feart.2020.571923, 2021

**Specific comments:**

There are some minor edits in the attached PDF.

Our point-by-point responses are attached below in blue, while the original reviewers' comments are in black.

Line 102: GPR data are critical for vslidation of thickness. Please add a section to describe GPR data, their sources, acquisition, distribution, and accuracy...

As response above, the GPR data used in this study is acquired from Azam et al. (2017), and this data is now available at Farinotti et al. (2021). Detailed information about GPR data is introduced by Azam et al. (2017). We add one sentence in Section 2.3.

*"The GPR data were measured based on a pulse radar system in October 2009 (Azam et al., 2017) and is available at Farinotti et al. (2021)."*

Line 105: is
Corrected.

Line 130: Delete "optics or SAR"
Done.

Line 161: Elevations ... are ... Elevations of other four ... are ...
Corrected.

Line 229: Change "exactly know" to "estimate".
Corrected.

Line 232: Change it to "4 sigma"
Corrected.

Line 280: Be consistent with "~" and "?+-".

It should be -1 to 1 m, we have corrected it.

Line 405: Please add above conditions.

Corrected.

*"Therefore, we conclude that the seasonal fluctuations of ICESat-2 data have little influence on the assessments of the DEMs under the above conditions."*

Line 449: Grid spacing. Same for the later cases.

Corrected.

**Reviewer #2**

The manuscript has considerably improved. Still many small details remain to be fixed, most of them related to grammar and spelling. All my comments are included in the annotated pdf. In general I think the results are relevant and the approach is sound, and I therefore recommend publication after minor revisions.

We thank the reviewer for the positive evaluation of our manuscript.

Line 21: increased -> increase

Corrected.

Line 23: Then -> In a next step,

Corrected.

Line 25: It is unclear how the second part of the sentence connects to the first. Please reformulate or just remove the first part of the sentence.

We have removed the first part of the sentence.

Line 25: "Our assessment first figures out the performances of mainly global DEMs over the glacierized TP." This may be removed. It is already clear from the earlier part of the abstract.

We have removed this sentence.

Line 27: Also this is rather obvious and may be deleted.

We agree with the suggestions from the reviewer and deleted this sentence.

Line 52: Overestimate -> overestimation

Corrected.

Line 53: No need to mention this, it is obvious from the previous.

We agree with the suggestions from the reviewer and deleted this sentence.

Line 54: Is this with GlabTop or another model? Please specify.

*We have added* *"... using Huss-Farinotti model (Huss and Farinotti, 2012)."*

Line 55: Not sure what is meant here. Please clarify or remove.
We have removed this sentence.

Line 55: most -> several, has -> have
Corrected.

Line 70: "and complicated" delete
Deleted

Line 73: DEM with high resolution is also of limitation -> high resolution DEMs are of lower quality
Corrected.

Line 74: DEM -> DEMs
Corrected.

Line 76: public freely-accessed -> open-access
Corrected.

Line 79: their studies -> Liu et al. (2019)
Corrected.

Line 103: Descriptions of ICESat-2 elevation data referenced -> ICESat-2 elevation data
Corrected.

Line 108: ICESat-2 -> ICESat-2's
Corrected.

Line 105: product both can -> products can both
Corrected.

Line 109-110: TL03 -> The ATL03
Corrected.

Line 110: considering -> based on
Corrected.

Line 110: Compute -> Computational
Corrected.

Line 127: Descriptions of global-scale DEMs evaluated -> Global DEMs
Corrected.

Line 129: evaluating their influences on ITIMs -> evaluation of ITIM sensitivity
Corrected.

Line 141: and the influence of ablation and accumulation of glaciers should also be noted. ->
inducing errors due to (seasonal and long-term) accumulation / ablation on glaciers.
Corrected.

Line 160: method -> methods
Corrected.

Line 170: Some more information or a reference for the GPR data would be useful.
We have added one sentence and one reference here.
*"The GPR data were measured based on a pulse radar system in October 2009 (Azam et al., 2017)
and is available at Farinotti et al. (2021)."*
Azam, M.F., Wagnon, P., Ramanathan, A., Vincent, C., Sharma, P., Arnaud, Y., Linda, A.,
    Pottakkal, J.G., Chevallier, P., Singh, V.B., & Berthier, E: From balance to imbalance: a shift in
    the dynamic behaviour of Chhota Shigri glacier, western Himalaya, India. J GLACIOL, 58,
    315-324, https://doi.org/10.3189/2012JoG11J123,2012

Line 171: ":"-> "."
Corrected.

Line 172: Start new paragraph here.
Corrected.

Line 173: bottom -> basal
Corrected.

Line 180: "The"
Included.

Line 214: This should be in bold text.
Corrected.

Line 215: measurement -> measurements
Corrected.

Line 216: outside -> greater than
Corrected.

Line 218: assessments -> analysis
Corrected.

Line 222: change ranged between

Corrected.

Line 223: acquiring -> acquisition, DEM -> DEMs
Corrected.

Line 223: error due to the glacier dynamic -> errors due to the changing glacier geometry.
Corrected.

Line 224: "TanDEM-X and AW3D30 are acquired in different months and years (Table 1), it's hard to analyse the impact of glacier dynamic on accuracy assessment" -> "This applies in particular to TanDEM-X and AW3D30, which are collected in different months and years."
Corrected.

Line 226-229: "The way this is formulated is confusing. In case the difference between ICESat-2 data and DEMs is used to correct the ICESat-2 data then the errors would reduce to zero when comparing DEMs and corrected ICESat-2 data. But I suppose that is not what is meant here. Please clarify what adjusted ICESat-2 means."
The adjusted ICESat-2 means the ICESat-2 elevation plus the glacier elevation change magnitude during 2000-2020. We have improved this sentence.
*"Then the glacier elevation dynamic magnitude during February 2000 and February 2019/2020 (Shean et al., 2020), are subtracted from the selected ICESat-2 elevation. By comparing the elevation differences from the adjusted ICESat-2 and the four DEMs, we could partly estimate the impacts of glacier dynamic on accuracy assessment."*

Line 233: ration- ratio
Corrected.

Line 246: differed little -> did not differ much
Corrected.

Line 254: It seems like that nearly all dashed lines (showing mean elevation difference) are too far to the right, i.e. that more than 50% of the area under the graph is on the left side of the dashed line. I could be wrong... Are the mean differences here corrected for glacier height change between acquisition dates or not (I was a bit confused by the description in the methods on this)?
In fact, we are aware that the elevation difference is affected by the glacier elevation change. The mean differences here are not corrected for glacier height change between acquisition dates. However, in the Discussion section, we estimated the influence from glacier dynamic on accuracy assessment.

Line 254: One dashed line has the wrong color.
Corrected.

Line 262: the nearly same -> nearly the same
Corrected.

Line 266: Thereinto?

Deleted.

Line 267: minimum -> smallest

Corrected.

Line 267: This sentence is not clear in its current form

This sentence was improved.

*"The STD of NASADEM was improved over the most part of TP, compared with that of SRTM-GL1 (Fig.5b)"*

Line 268: indicate -> indicates

Corrected.

Line 270: improvement (?) Compared to what exactly?

The sentence is reformulated.

*"...large improvements in STD..."*

Line 271: are almost same in space -> have the same spatial distribution

Corrected.

Line 271: "corresponding to their" -> "and have"

Corrected.

Line 279: amplitude -> amplitude is found

Corrected.

Line 285: serious -> steep

Corrected.

Line 288: behave best against the slope -> perform best for steep slopes

Corrected.

Line 292: show the similar -> show a similar

Corrected.

Line 295: high elevation region -> highest elevations

Corrected.

Line 300: Differences -> Elevations differences

Corrected.

Line 314-315: It could help to include the acquisition dates somehow in Fig. 7. That makes the

figure much easier to interpret.

We have improved Fig.7 a-d.

[Figure]

Line 324: The DEMs do not "model" the ice thickness, please rephrase, e.g. "Sensitivity of modelled ice thickness to DEMs"

We have changed the title of Section 3.4 to "Sensitivity of modelled ice thickness to DEMs"

Line 327: on specific ITIMS -> "on ice thickness estimated using different ITIMs"
Corrected.

Line 332: from -> when
Corrected.

Line 333: maximal -> largest
Corrected.

Line 334: Minimum -> smallest
Corrected.

Line 339: weakness -> weaknesses
Corrected.

Line 339: 8, 7, 3 and 2 what?
We have improved this sentence.
*"NASADEM indicates better performance (number of smaller RMSE in five profiles) relative to other DEMs by using GlabTop2 and ITBIOV models. AW3D30 conducts better by using ITBIOV and OGGM models. TanDEM-X is better by using OGGM model. While five models are composited, NASADEM behaves better."*

Line 350: The benefit of using all models (weighted) instead of one model could be quantified, i.e. how much smaller is the RMSE for all models combined when compared to the RMSE for one ITIM? The values are already in Table 2.
We add one sentence.
"RMSEs of combined ice thickness modelled from different DEMs are reduced by ~21 m (~25%), when compared to the RMSE for one ITIM."

Line 370: in -> near
Corrected.

Line 390: So the results presented so far have not been corrected for different acquisition dates and related elevation change, right? It could be good to make this more clear earlier on in the manuscript.
Yes, the results presented so far are not corrected for different acquisition dates and related elevation change as we cannot acquire the exact date of TanDEM-X and AW3D30. We state this information in Method section.
*"Then the glacier elevation dynamic magnitude during February 2000 and February 2019/2020 (Shean et al., 2020), are subtracted from the selected ICESat-2 elevation. By comparing the elevation differences from the adjusted ICESat-2 and the four DEMs, we could partly estimate the impacts of glacier dynamic on accuracy assessment."*

Line 398: "the glacier melt and sublimate" -> "glaciers experience more melt and sublimation" ?
Corrected.

Line 399: "the glacier"-> ". As a result, the"
Corrected.

Line 399: ";then" -> "and"
Corrected.

Line 400: "fewer" -> less
Corrected.

Line 414: have a strong dependence -> depend strongly

Corrected.

Line 415: Not sure what is meant here... Why would the probability density of aspects matter? Please reformulate or remove.

We have removed this sentence.

Line 419: accordant distribution of data in different slopes with aspect -> predominance of slopes for certain aspects.

Corrected.

Line 435: serious -> substantial, with the output -> and the output appears

Corrected.

Line 440: "Pixels in DEMs do not represent exactly the same location." (?)

You are right, and we have corrected this sentence.

Line 447: No need to mention MATLAB here, but rather indicate what type of fit function / interpolation technique is used.

We have removed "in MATLAB", and added "nonlinear least squares" in this sentence.

Line 449: offset pixels -> misregistration

Corrected.

Line 449: are all at -> is always

Corrected.

Line 457: model -> ITIM

Corrected.

Line 457: have different outcomes -> will yield different thickness patterns

Corrected.

Line 458: uncertainty of DEM -> quality of a DEM

Corrected.

Line 459: maximal and minimum -> largest and smallest

Corrected.

Line 462-466: This is rather Results than Discussion. Furthermore, the way in which the perturbations are applied is not described if I am correct. It should be made clear whether perturbations are applied everywhere as a systematic perturbation or as random noise. Maybe I missed it...

Here, we described the sensitivity of GlabTop2 and ITIBOV models to slope and elevation. It could be better to be put in Discussion section. We have added one sentence to describe the sensitivity test.

*"A sensitivity test based on the Equations 1−4 was executed and the modelled ice thickness differences before and after adding the input parameters such as slope and elevation were compared."*

Line 469: Is it ITIBOV or ITBOV? Please check consistency throughout the manuscript
It's ITIBOV, we have checked throughout the manuscript.

Line 471: rather use "ice thickness" instead of "output"

Line 472: "the" delete
Corrected.

Line 475: "motion" -> "transient dynamics"
Corrected.

Line 476: "When the better accuracy of" -> "With the higher-accuracy"
Corrected.

Line 477: "led to the relatively best outcomes" -> most accurate ice thicknesses were obtained
Corrected.

Line 478: "results"-> "ice thicknesses"
Corrected.

Line 479: "30-m DEMs comparied with 90-m resolution DEMs" -> "using 30 m or 90 m DEMs as input."
Corrected.

Line 479: "means that high spatial resolution improved the outcome little"-> thereby suggesting a minor impact of DEM resolution on ice thickness reconstruction
Corrected.

Line 479: "For" -> "In"
Corrected.

Line 481: "For" -> "In"
Corrected.

Line 486: models->ITIMs
Corrected.

Line 487: not sure what is meant with this
We have improved this sentence.
*"When the results from different ITIMs are ensembled, the influences of uncertainty and resolution*

*in the input DEMs on the modelled ice thickness still exist (Fig. 9 and Table 2)."*

Line 488: equipped with -> which have
Corrected.

Line 488: achieved -> yielded
Corrected.

Line 488: "best outcomes" -> "most accurate thickness estimates"
Corrected.

Line 490: the discord -> "spatial inconsistencies(?)"
Corrected.

Line 496: what's more -> "Furthermore,"
Corrected.

Line 497: "as similar as" -> ", similar to what has been done in"
Corrected.

Line 526: best outcomes -> "most accurate thickness estimates"
Corrected.

Line 529: limited by -> "but with limitations from"
Corrected.